

# Mobile and high spectral resolution Fabry Pérot interferometer spectrographs for atmospheric remote sensing

Jonas Kuhn[1,2], Nicole Bobrowski[1,2], Thomas Wagner[2], and Ulrich Platt[1,2]

[1]Institute of Environmental Physics, University of Heidelberg, Heidelberg, Germany
[2]Max Planck Insitute for Chemistry, Mainz, Germany

**Correspondence:** Jonas Kuhn (jkuhn@iup.uni-heidelberg.de)

**Abstract.**

Grating spectrographs (GS) are presently widely in use for atmospheric trace gas remote sensing in the ultraviolet (UV) and visible spectral range (e.g. differential optical absorption spectroscopy, DOAS). For typical DOAS applications, GSs have a spectral resolution of about half a $nm$ corresponding to a resolving power $R$ (ratio of operating wavelength to spectral resolu-
tion) in the range of 1000. This is sufficient to quantify the vibro-electronic spectral structure of the absorption of many trace gases with good accuracy and further allows for mobile (i.e. compact and stable) instrumentation.

However, a much higher resolving power ($R \approx 10^5$, i.e. a spectral resolution of about the width of an individual rotational absorption line) would facilitate the measurement of further trace gases (e.g. OH radicals), significantly reduce cross interferences due to other absorption and scattering processes, and provide enhanced sensitivity. Despite of these major advantages,
only very few atmospheric studies with high resolution GSs are reported, mostly because increasing the resolving power of a GS leads to largely reduced light throughput and mobility. However, for many environmental studies, light throughput and mobility of measurement equipment are central limiting factors, for instance when absorption spectroscopy is applied to quantify reactive trace gases in remote areas (e.g. volcanoes) or from air borne or space borne platforms.

Since more than a century, Fabry Pérot interferometers (FPIs) have been successfully used for high resolution spectroscopy
in many scientific fields where they are known for their superior light throughput. However, except for a few studies, FPIs received hardly any attention in atmospheric trace gas remote sensing, despite their advantages. We propose different high resolution FPI spectrograph implementations and compare their light throughput and mobility to GSs with the same resolving power. We find that nowadays mobile high resolution FPI spectrographs can have a more than two orders of magnitude higher light throughput than their immobile high resolution GS counterparts. Compared to moderate resolution GSs (as routinely used
for DOAS), a FPI spectrograph reaches a 250 times higher spectral resolution while the signal to noise ratio (SNR) is reduced by only a factor of 10. With a first compact prototype of a high resolution FPI spectrograph ($R \approx 148000$, <8 litres, <5 kg) we demonstrate that these expectations are realistic.

Using mobile and high resolution FPI spectrographs could have a large impact on atmospheric near UV to near IR remote sensing. Applications include the enhancement of sensitivity and selectivity of absorption measurements of many atmospheric
trace gases and their isotopes, the direct quantification of OH radicals in the troposphere, high resolution $O_2$ measurements for radiative transfer and aerosol studies and solar induced chlorophyll fluorescence quantification using Fraunhofer lines.




# 1 Introduction

The Fabry-Pérot interferometer (FPI) was introduced at the end of the 19th century and has since then led to a tremendous progress in many areas of spectroscopy (as summarized in e.g. Vaughan, 1989). For resolving powers ($R = \frac{\lambda}{\delta\lambda}$) higher than a

few thousands, Jacquinot (1954, 1960) showed that the FPI exhibits a fundamental luminosity (or light throughput) advantage over gratings. In that time and until the 1970s, most spectrometers were implemented as a scanning monochromator using a one-pixel detector (e.g. a photomultiplier tube). The luminosity advantages were, however, also found for the - in that time so-called - "photographic use" of a spectrometer (i.e. a spectrograph), where photographical plates were used as focal plane detector.

Nowadays, grating spectrographs (GSs) with one - or two - dimensional detector arrays (e.g. CCD or CMOS detectors) are widely used for atmospheric remote sensing of trace gases in the near UV to near IR spectral region (see Platt and Stutz, 2008). Even when scattered sunlight is used as lightsource, they offer sufficient signal to noise ratios (SNRs) for moderate resolving powers ($R \approx 1000$) and compact and stable (i.e. mobile) instrumentation without moving parts.

Despite of substantial benefits of increased spectral resolution for noumerous atmospheric remote sensing applications (see

below), the advantages of FPIs are widely ignored, probably for the following major reasons: (1) Many trace gases can be detected with moderate resolving power due to moderate resolution (vibro-electronic) absorption structures in the UV and visible spectral range. (2) In contrast to FPI spectrographs, GSs are commercially readily available and relatively affordable. (3) For broadband light sources (as the case in many atmospheric measurements) FPIs require further optical components for order sorting. (4) Furthermore, as concluded by Jacquinot (1960), FPIs "will probably always suffer from the fact that the

dispersion is not linear".

In this work, we show that it is worthwhile to consider the use of FPIs in spectrographs for remote sensing measurements in the atmosphere. Detection limits of many trace gases can be lowered by orders of magnitude, while maintaining mobile instrumentation.

First, we discuss the benefits of high resolution atmospheric trace gas remote sensing and introduce some past applications

and their limitations (Sect. 1.2). Then, basic aspects of mobility are briefly introduced (Sect. 1.3). In Sect. 2, we sketch high resolution FPI spectrograph designs that can be implemented in mobile and stable instruments. In Sect. 3, the luminosity and physical size of the proposed FPI spectrograph implementations are compared to a GS with the same resolving power. By scaling the GSs performance, SNRs of known moderate resolution atmospheric measurements are used to anticipate SNRs for the proposed FPI spectrographs. Extensive details of those calculations as well as lists of symbols and abbreviations are

presented in the appendices. In Sect. 4, we discuss the results regarding the potential impact of FPI spectrograph technology on atmospheric sciences and, finally, introduce a first prototype of a FPI spectrograph.

## 1.1 Definitions and conventions

Throughout the paper we use spectroscopic terminology, which might have slightly varying meanings in different fields of spectroscopy. To avoid confusion they are briefly explained here.



A spectrograph is a spectrometer where the components of the spectrum are separated in space and recorded simultaneously with a detector array. The instrument line function (ILF) $H$ describes the response of a spectrograph to an input of spectrally infinitesimal width (i.e. monochromatic radiation). The ILF determines the spectral interval that can be resolved by the spectrograph. This interval is in the following called a spectral channel of the spectrograph (not to be confused with the spectral range covered by a pixel of the spectrograph's detector). Its full width at half maximum (FWHM, denoted by $\delta\lambda$) can be used

(amongst other and rather similar definitions) to quantify the spectral resolution. What we call a high spectral resolution corresponds to a narrow width of a spectral channel (i.e. a low value of $\delta\lambda$). The spectral range covered by all spectral channels of the spectrograph describes its spectral coverage. The resolving power $R$ of the spectrograph is the ratio of operating wavelength $\lambda$ and spectral resolution $\delta\lambda$. Investigating the light throughput of spectrographs on a spectral channel basis allows the direct comparison of their noise limited detection limits for trace gas absorption (see Sect. 3).

In spectroscopic atmospheric trace gas remote sensing the column density $S$ of the gas is directly quantified. The column density denotes the concentration of the trace gas integrated along the respective measurement light path. According to different experiment desings and applications the light path differs and finally determines the detection limit in terms of concentration (see e.g. Platt and Stutz, 2008, for details).

## 1.2 Atmospheric trace gas remote sensing with high spectral resolution

The width of rovibronic absorption lines of atmospheric trace gas molecules in the near UV to near IR spectral range, as well as many Fraunhofer lines are on the order of some pm. In order to observe the corresponding spectral structures (in particular individual rotational lines), resolving powers in the range of $R \approx 10^5$ are required. This defines what we refer to in the following as "high spectral resolution".

In the UV and visible spectral range many trace gas molecules show 'bands' of absorption lines composed of many, partially
overlapping rotational lines of a vibrational transition, resulting in structured absorption cross sections, even when observed with moderate spectral resolution ($R \approx 1000$). These trace gas molecules can be quantified along light paths inside Earth's atmosphere by differential optical absorption spectroscopy (DOAS, see Platt and Stutz, 2008). Compact moderate spectral resolution GSs are used to record spectra of direct or scattered sun light or artificial light sources from ground based to space borne platforms and thereby allow for spatially and temporally resolved measurements of (also very reactive) trace gases.

However, a higher spectral resolution is desirable in many cases. There are atmospheric trace gases that are more difficult or even impossible to measure with moderate resolution. For instance hydroxyl radicals (OH) exhibit distinct and narrow absorption lines (widths of 1-2 pm at 308 nm, see Fig. 1). Due to the low atmospheric concentrations, the absorption can not be separated from overlaying effects (e.g. other absorbing gases) with spectral resolutions that are much lower than the width of the individual lines. Tropospheric OH concentrations were measured with high resolution absorption spectroscopy
by e.g. Perner et al. (1976); Platt et al. (1988) or Dorn et al. (1996) using large GS setups (850-1500 mm focal length) and an intricate broad band laser system as light source (as described in Hübler et al., 1984). Direct sun light measurements of OH were performed with Fourier Transform Spectroscopy (FTS, e.g. Notholt et al., 1997), high resolution GS (1500 mm focal length, Iwagami et al., 1995) and rather delicate systems employing series of pressure tuned FPIs (e.g. Burnett and Burnett,





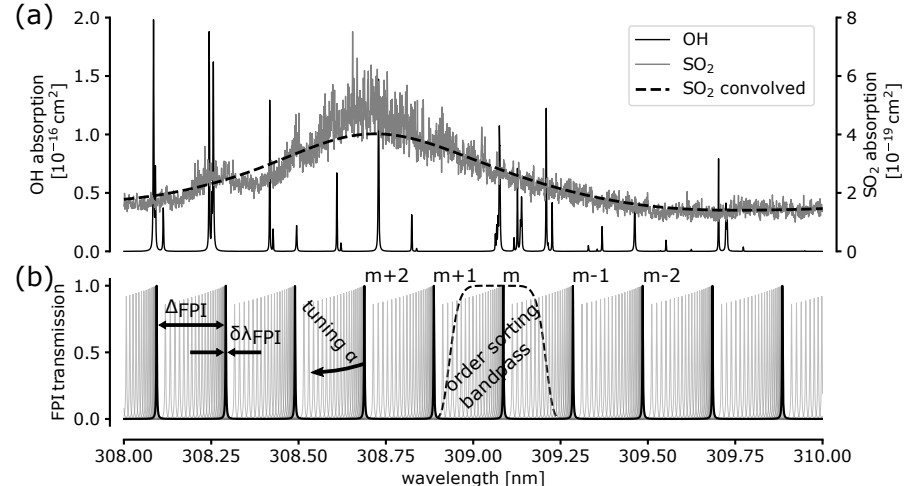

**Figure 1.** (a) OH absorption cross section (Rothman et al., 2013, left ordinate axis) and $SO_2$ absorption cross section by (Rufus et al., 2003, right ordinate axis). The dashed line shows a convolution of the $SO_2$ absorption with a Gaussian of 0.4 nm width. (b) FPI transmission spectrum (black drawn line) as it is scanning across a short wavelength range (indicated by the gray lines) by tuning of an instrument parameter (here the incidence angle $\alpha$ is tuned from $0°$ to $2°$ in $0.05°$ steps). The decrease in peak transmission is due to an assumed small beam divergence ($0.005°$ half opening angle). An order sorting bandpass is indicated by dashed line isolating the FPI peak of order $m$.

1981). Further, high resolution $O_2$ measurements were performed in the atmosphere by e.g. Pfeilsticker et al. (1998), using
a GS (1500 mm focal length). The high spectral resolution allows to quantify the absorption of individual lines of different strength and thereby to infer for instance light path length distribtuions in clouds. The rather complex and immobile hardware of the named measurements limited their application to a few and locally restricted atmospheric studies.

  Many other atmospheric trace gases show strong and structured absorption on the pm scale. Besides sulfur dioxide ($SO_2$, e.g. Rufus et al., 2003), formaldehyde (HCHO, e.g. Ernest et al., 2012) water (Rothman et al., 2013), and chlorine monoxide (ClO
Barton et al., 1984), Neuroth et al. (1991) found strong, discrete and narrow bromine monoxide (BrO) absorption lines in the UV. Using these much more detailed and specific spectral features of the trace gases could not only substantially increase the selectivity but also in many cases increase the sensitivity of DOAS measurements. Additionally, the absorption cross sections of isotopes of some trace gases could be distinguished, similar to the moderate spectral resolution measurements of water vapour isotopes (e.g. Frankenberg et al., 2009). Figure 1a illustrates the addressed difference in spectral resolution by showing
the high resolution absorption cross section of $SO_2$ (Rufus et al., 2003) together with a convolution representing the absorption cross section as seen by a compact GS with 0.4 nm spectral resolution.

Moderate resolution scattered sunlight DOAS measurements largely undersample solar Fraunhofer lines (the width of which can also be in the pm range). This on the one hand introduces uncertainties in the effective spectral absorption of the trace gases (see e.g. Lampel et al., 2017). On the other hand, in most cases it implies the need for a Fraunhofer reference spectrum. High
resolution spectra would allow a direct separation of Fraunhofer structures from narrow trace gas absorption structures and,





moreover, absolute atmospheric column densities of trace gases could be determined (rather than the column density relative to a reference spectrum).

### 1.3 Instrument mobility

A key point of the success of moderate spectral resolution DOAS measurements in the atmosphere is the use of compact and
stable (i.e. mobile) spectrographs (volume of the order of $1\,\mathrm{litre}$ with a focal length $f$ of about $10\,\mathrm{cm}$, no moving parts). As mentioned above they typically yield a resolving power in the range of 1000 and a light throughput that allows the recording of scattered sunlight spectra in the UV and visible spectral range with a SNR of several thousands within less than a minute (e.g. Lauster et al., 2021). This is sufficient to retrieve many of the weakly absorbing atmospheric trace gases in the UV and visible spectral range (optical densities of ca. 0.01 - 0.0001) and to study their dynamics and chemistry.

Mobility of measurement equipment provides substantial advantages to practical field applications: (1) deployment on mobile platforms like e.g. cars, camels, drones, balloons, aircrafts and miniature satellites. (2) Significant reduction of costs for field campaigns due to reduced infrastructure and human resource requirements. (3) Remote locations, e.g. deserts or volcanic craters are made accessible with e.g. backpack sized instruments. (4) Instruments can be employed in autonomous, remote and low-maintenance measurement networks (see e.g. Galle et al., 2010; Arellano et al., 2021). In practice, these points are
substantial factors making scientific environmental observations feasible.

As will be shown below, increasing the resolving power of a GS also requires a larger instrument size. Thereby the mobility advantages are largely lost. The use of FPIs in spectrograph setups can yield high resolving powers while maintaining a high mobility of the instrument.

### 1.4 Fourier transform spectroscopy

This work focuses on spectrograph setups (GS, FPI spectrographs) because of their high stability (no movable parts) and low sensitivity to fluctuations in light intensity. FTSs (i.e. Michelson interferometers) do not fulfill these requirements. A one-pixel detector records interferograms in a temporal sequence while mechanical changes in the optics (i.e. the interferometer path length) are conducted. This already imposes limitations to FTS's mobility and its applicability to more dynamical measurement conditions (e.g. cloudy skies). Since, besides GSs, FTSs are in broader use in atmospheric remote sensing (mostly towards
longer wavelengths, where the well established and cost effective technology of silicon detector arrays can not be used anymore, i.e. above ca. $1100\,\mathrm{nm}$) they shall nevertheless be shortly mentioned here.

In contrast to GSs and FPI spectrographs FTSs reach a large spectral coverage with very high and adjustable spectral resolution. This can be an important advantage for many atmospheric studies.

Notholt et al. (1997) compared the SNR of high resolution ($R \approx 300000$) FTS measurements with that of GS measurements
with similar resolving power of Iwagami et al. (1995) for direct sunlight measurements at around $308\,\mathrm{nm}$. It was found that the SNR of the FTS and GS were similar for clear sky conditions and worse for the FTS under hazy or slightly cloudy conditions. Thus the advantages of FPI spectrographs regarding the SNR found below (see Sect. 3.2.3) are expected to similarly hold for



a FPI spectrograph to FTS comparison. While the spectral coverage for high resolution of FTSs is superior, mobility aspects (movable parts and large focal lengths in FTSs) clearly favour FPI spectrographs.

## 2 High resolution spectroscopy with Fabry Pérot-interferometers

FPIs are very simple optical instruments that are known for a long time. However, progress in manufacturing processes led to largely improved properties in the last decades. A FPI consists of two plane-parallel reflective surfaces (mirrors, see Fig. 2a). Since incident light is reflected back and fourth between these surfaces, interference of transmitted and reflected partial beams leads to spectral transmission patterns determined by the optical path length between the two surfaces (see e.g. Perot and Fabry, 1899; Vaughan, 1989, for details). This optical path length and the optical path difference $\Gamma$ is determined by the physical separation of the reflective surfaces $d$, the refractive index $n$ of the medium between the surfaces and the angle of incidence $\alpha$ of the incoming light:

$$\Gamma = 2\,d\,n\,\cos\alpha \tag{1}$$

Thus, the transmission maximum (constructive interference) with the order $m$ is centered at the wavelength

$$\lambda_{\mathrm{m}} = \frac{\Gamma}{m} \tag{2}$$

The free spectral range (FSR) $\Delta\lambda_{\mathrm{FPI}}$ describes the spectral separation of two neighbouring transmission peaks (or fringes) and is related to a transmission peak's FWHM $\delta\lambda_{\mathrm{FPI}}$ via the finesse $\mathcal{F}$ (see Fig. 1b):

$$\Delta\lambda_{\mathrm{FPI}} = \mathcal{F}\,\delta\lambda_{\mathrm{FPI}} \approx \frac{\lambda^2}{\Gamma} \tag{3}$$

The spectral resolution of a FPI transmission order or in other words the spectral width of its ILF is thus given by $\delta\lambda_{\mathrm{FPI}}$. The isolation of a single FPI peak is desired for broadband light sources, unless correlation of FPI transmission spectrum with trace gas spectrum can be exploited (as e.g. in Vargas-Rodríguez and Rutt, 2009; Kuhn et al., 2014, 2019). An order sorting bandpass (i.e. the isolation of a wavelength range containing a single FPI fringe, see Fig. 1b) can be achieved by a bandpass filter, further FPIs (or a combination of both, see e.g. Mack et al., 1963) or dispersive elements like a grating or a prism (e.g. Fabry and Buisson, 1908). The order sorting bandpass needs to be in the range of the FSR of the FPI. Through Eq. (3) the spectral resolution $\delta\lambda_{\mathrm{FPI}}$ of a FPI spectrograph is thus limited by the FPI's finesse and the order sorting bandpass. The finesse of a FPI indicates the number of interfering partial beams and, thus, depends on the reflectivity, the alignment and the quality of the FPI mirror surfaces across its clear aperture (CA, e.g. diameter of usable circular aperture). Therefore, it is to a large extent limited by the manufacturing process. Nowadays, high finesse across larger CAs is reached by static, air-spaced FPI setups, i.e. FPIs with fixed $d$ and low thermal expansion glass spacers. The spectral width of bandpass filters, which in principle also consist of a sequence of interference layers, are limited by manufacturing processes in a similar way. Thus, the measurement application and the available optical components determine the appropriate order sorting technique.

In order to resolve different wavelengths, the FPI has to be operated in a range of varied physical parameters ($d$, $n$ or $\alpha$),



resulting in a spectral shift of the FPI transmission (as indicated in Fig. 1b). This can be implemented in different ways (see e.g. Vaughan, 1989). For high finesse FPIs, pressure or temperature tuning (i.e. changing the refractive index $n$ of the medium

between the mirrors) or using the dependence on the incidence angle $\alpha$ are preferred. The variation of the mirror separation $d$ across the FPIs CA often limits the finesse by impacting the parallelism of the mirrors. An extremely precise tuning of $d$ would be required. Pressure tuning requires for instance to ramp the pressure inside the FPI. While this can only be done in a time sequence, the use of detector arrays allows for observing different incidence angles $\alpha$ simultaneously in spectrograph implementations without moving parts. For studying dynamic processes in the atmosphere and also for optimised mobility a

static spectrograph setup is highly preferred.

In the following, sample calculations are mostly made for short wavelengths ($\approx 300\,\mathrm{nm}$), where FPI manufacturing is most challenging. For increasing wavelengths the inferred performance tends to improve since the absolute finesse limiting requirements concerning roughness, parallelism or sphericity of the mirror surfaces (often given as fraction of wavelength, e.g. $\lambda/100$) are higher for lower wavelengths.

## 2.1 FPI spectrograph implementation for atmospheric remote sensing

A simple and compact FPI spectrograph can be implemented with a static FPI and an optics that images the different FPI incidence angles of the traversing light beam to concentric rings of equal FPI transmission on the focal plane (see Fig. 2a). There, a detector array records the intensities of the different spectral channels simultaneously. The spectral shift of the FPI transmission due to a small change of the small incidence angle $\alpha$ (i.e. a few hundredth of a radian, $\alpha \approx \sin\alpha \approx \tan\alpha$, $\cos\alpha \approx$

1) is dependent on the wavelength $\lambda_{\mathrm{m}}$ of the transmission peak of the order $m$ and $\alpha$ itself (see Eq. (1) and (2)):

$$\frac{\mathrm{d}\lambda_{\mathrm{m}}}{\mathrm{d}\alpha} = \frac{2\,d\,n}{m}\frac{\mathrm{d}}{\mathrm{d}\alpha}\cos\alpha = \frac{-2\,d\,n}{m}\sin\alpha \approx -\lambda_{\mathrm{m}}\,\alpha \tag{4}$$

This demonstrates the non-linearity of the dispersion, which, however, leads to a constant light throughput for all spectral channels (as described in detail below). The wavelength range $\Lambda_{\mathrm{m}}$ covered by a particular transmission order, i.e. the FPI's spectral tuning range, is determined by the angle range covered by the parallelised light beam traversing the FPI:

$$\Lambda_{\mathrm{m}} = -\lambda_{\mathrm{m}}\int\limits_{|\alpha_{\mathrm{min}}|}^{|\alpha_{\mathrm{max}}|}\mathrm{d}\alpha\,\alpha \tag{5}$$

The maximum and minimum incidence angles $\alpha_{\mathrm{max}}$ and $\alpha_{\mathrm{min}}$ are determined by the illuminated entrance aperture $B$ and the focal length of lens 1 (collimating lens) of the FPI spectrograph's imaging optics (see Fig. 2a). For the imaging axis centered at the optical axis ($\alpha_{\mathrm{min}} = 0$) the maximum incidence angle is:

$$\alpha_{\mathrm{max}} \approx \frac{B}{2\,f_1} \tag{6}$$

Assuming for instance an entrance aperture of $B = 3\,\mathrm{mm}$ and a focal length $f_1 = 50\,\mathrm{mm}$ the maximum FPI incidence angle would be $\alpha_{\mathrm{max}} = 0.03$ (or $1.72°$) and the spectral coverage about $0.135\,\mathrm{nm}$ at $300\,\mathrm{nm}$. The practical incidence angle range that can be imaged onto the focal plane is in the range of a few degrees and therefore the wavelength coverage typically can



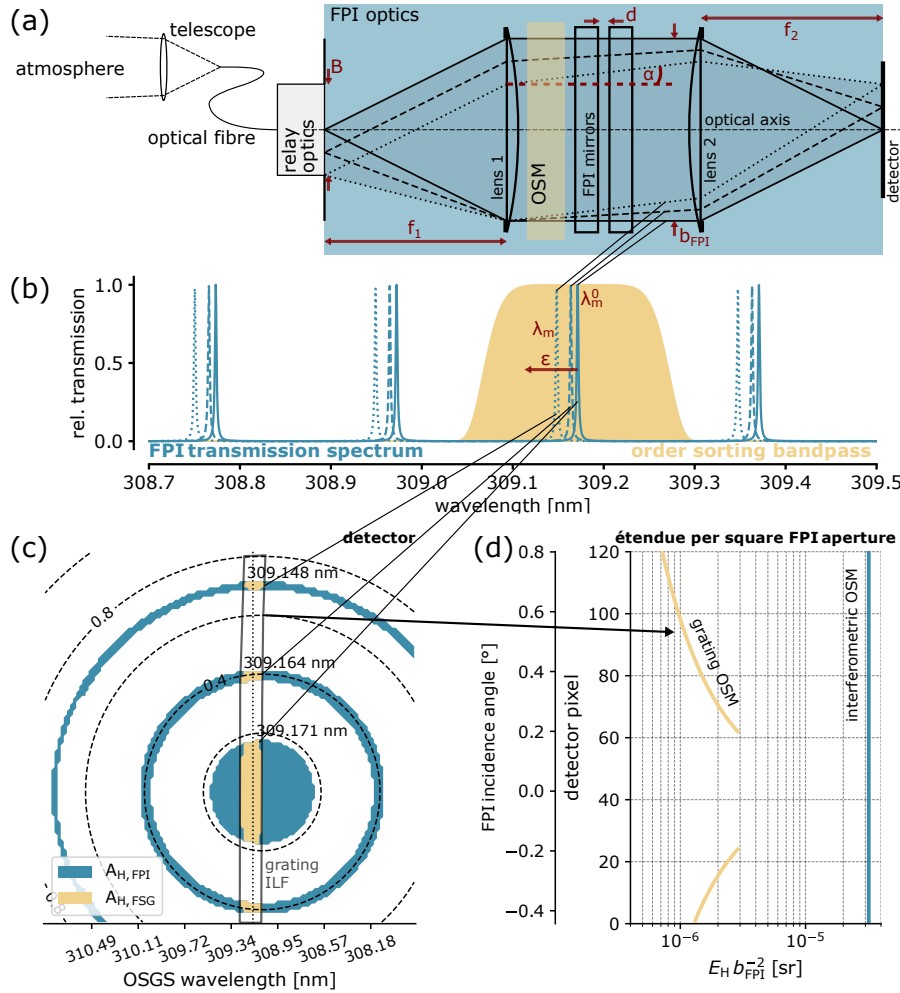

**Figure 2.** Schematic optical setup of an FPI spectrograph: (a) Light from the atmosphere is directed to the spectrograph entrance via a telescope, an optical fibre and, if needed, a relay optics. The order sorting mechanism (OSM) depending on its implementation can be at different locations within the optical path. Lens 2 images the different FPI incidence angles onto the image plane, thereby different spectral FPI transmission spectra (b) are separated on the focal plane detector (c). The dashed circles in panel (c) indicate the corresponding FPI incidence angle $\alpha$ (in degree). The OSM isolates a single FPI transmission order, either via filters (interferometric) or via a grating (see grating ILF in (c)). (d) Étendue per square FPI aperture for the two OSMs and instrument parameters in Tab. 1.

reach some hundreds of pm in the near UV. A moderate resolution FPI spectrograph (with $R \approx 1000$, i.e. some hundreds of pm spectral resolution at 300 nm) of the proposed implementation would exhibit a spectral coverage on the order of its spectral

resolution, which would render it rather useless. This problem could be solved by tilting the FPI with respect to the imaging optical axis. Moderate resolution FPI spectrographs are, however, not addressed in this study. For a resolving power of about $10^5$ the 0.135 nm wavelength range at 300 nm would be divided in about 45 spectral channels with a 3 pm spectral resolution.





This is about the number of spectral channels used in a typical moderate resolution DOAS fitting window.

The sampling of the different spectral channels can be adjusted via the detector pixel size and the focal length of lens 2. Due to the non-linear dispersion the sampling needs to be adjusted to the outermost ring corresponding to the spectral channel with the lowest wavelength of a FPI order (when assuming equally sized pixels). For the above example ($B = 3\,\mathrm{mm}$, $f_1 = 50\,\mathrm{mm}$) and $f_2 = 50\,\mathrm{mm}$, the radial extension of the outermost spectral channel is about $\frac{\delta\lambda_{\mathrm{FPI}}}{\lambda\alpha} f_2 \approx 17\,\mu\mathrm{m}$ (see Eq. (4)). Nowadays, detector pixels with a 1-5 $\mu$m pitch are common. This would facilitate sufficient sampling ($> 3.4$ pixels per spectral channel width) for all spectral channels. The spectral sampling can further be adjusted via the focal length of lens 2. Since the intensities of all pixels with the same wavelength are co-added, this does not affect the light throughput.

The above mentioned order sorting mechanisms (OSMs) allow two basic FPI spectrograph implementations: (1) Using a grating as OSM in a FPI spectrograph results in a superposition of the linear grating dispersion with the radially symmetrical FPI transmission on the detector (see Fig. 2c). This allows for recording several FPI transmission orders at once and thereby increases the total spectral coverage of the FPI spectrograph. This OSM is referred to as grating OSM in the following.

(2) Using a combination of further FPIs and filters as OSM leads to optimised étendue for a wavelength coverage of a single transmission order but, thereby also to a reduced total wavelength coverage (only a single FPI order). This OSM is referred to as interferometric OSM in the following. As already mentioned above, the choice of the OSM depends on the measurement application, particularly the radiance of the light source, the desired SNR, the required spectral coverage and the manufacturability of optical components.

An optical fibre and, as the case requires, a relay optics direct the light collected by a telescope to the entrance aperture $B$ (see Fig. 2a). From there it traverses the imaging optics, containing the FPI together with the OSM (certainly, the OSM can also be in front or behind the FPI imaging optics, for instance the focal plane of an order sorting GS could be re-imaged).

Both OSM implementations allow simple, stable and mobile setups with no moving parts. Therefore they can be applied similarly to moderate resolution compact grating spectrographs in field measurement campaigns, autonomous measurement networks in remote areas, as well as in air borne or satellite applications.

## 3 Comparison of FPI spectrograph and GS

In this section, we compare the FPI spectrograph with the GS. First, size scaling considerations illustrate intrinsic mobility differences of FPI and grating instruments. Then, the light throughput per individual spectral channel is calculated and compared for different spectrograph implementations. From known SNRs of atmospheric measurements with moderate resolution GSs, the SNRs of the high resolution spectrographs can be approximated.

### 3.1 Fundamental differences and size considerations

When examining spectroscopic methods a basic question is how a physical parameter changes as a function of the wavelength $\lambda$. For spectrographs this physical parameter is most often a deflection angle $\theta(\lambda)$ of a light beam. The angular dispersion





describes the dependence of deflection angle $\theta_g(\lambda)$ on the wavelength for the grating. For the FPI spectrograph the incidence

angle dependence of the FPI transmission spectrum is used to separate the different spectral channels (see Sect. 2). Here, we

therefore regard the incidence angle $\alpha$ as equivalent deflection angle $\theta_{fp}(\lambda)$ for the FPI.

For a blazed grating with a given ruling distance $r_g$ operated in the $m$-th order and a Littrow type spectrograph setup (incidence

angle and dispersion angle as nearly equal as possible) the relation of wavelength and deflection angle $\theta_g$ (which in this case

equals the gratings blaze angle) is given by (see e.g. Jacquinot, 1954):

$$m\,\lambda = 2\,r_g \sin\theta_g \tag{7}$$

And consequently:

$$\frac{\mathrm{d}\lambda}{\mathrm{d}\theta_g} = \frac{2\,r_g}{m}\cos\theta_g \tag{8}$$

A close to ideal choice of the ruling distance of the grating for a given wavelength is $r_g \approx m\,\lambda$. For a typical value of $\theta_g = 30°$

a small wavelength shift by the width $\delta\lambda$ of one ILF (or one spectral channel) changes $\theta_g$ by

$$\delta\theta_g \approx 0.58\,\frac{\delta\lambda}{\lambda} \tag{9}$$

For the FPI the angle dependence (for small incidence angles) is given by Eq. (4) :

$$\frac{\mathrm{d}\lambda}{\mathrm{d}\theta_{fp}} = -\lambda\,\theta_{fp} \tag{10}$$

The same small wavelength shift by one spectral channel $\delta\lambda$ changes $\theta_{fp}$ by

$$\delta\theta_{fp} \approx \frac{1}{\theta_{fp}}\frac{\delta\lambda}{\lambda} \tag{11}$$

This means that the angular change $\delta\theta_g$ for a single spectral channel of the GS is approximately given by its inverse resolving

power, while for low FPI incidence angles the angular change $\delta\theta_{fp}$ for a wavelength change by $\delta\lambda$ can easily be two orders of

magnitude larger than its inverse resolving power (e.g. factor of 100 for $\theta_{fp} \approx 0.6°$).

In either type of spectrograph the angular deflection is translated to a spatial separation $\delta x$ on a detector array via an imaging

optics with focal length $f$ (see Fig. 3b or $f_2$ in Fig. 2a):

$$\delta x \approx f\,\delta\theta \tag{12}$$

The desired spatial interval per spectral channel on the detector depends on the pixel size and the spectral sampling. Assuming

the ILF to be sampled with five pixels of $10\,\mu m$ pitch, this interval would be $\delta x = 50\,\mu m$. Given that the size of a spectrograph

is of the order of its focal length and its volume and mass scale with its third power (see e.g. Platt et al., 2021), the above

relations reveal the principal difference of GS and FPI spectrograph in terms of size and resolving power (see Fig. 3a). For

instance, one could argue that easily portable tools for humans have the size of a human hand, i.e. ca. $10\,cm$, which at the

same time is about the size of a CubeSat miniature satellite (see e.g. Poghosyan and Golkar, 2017). The resolving power of

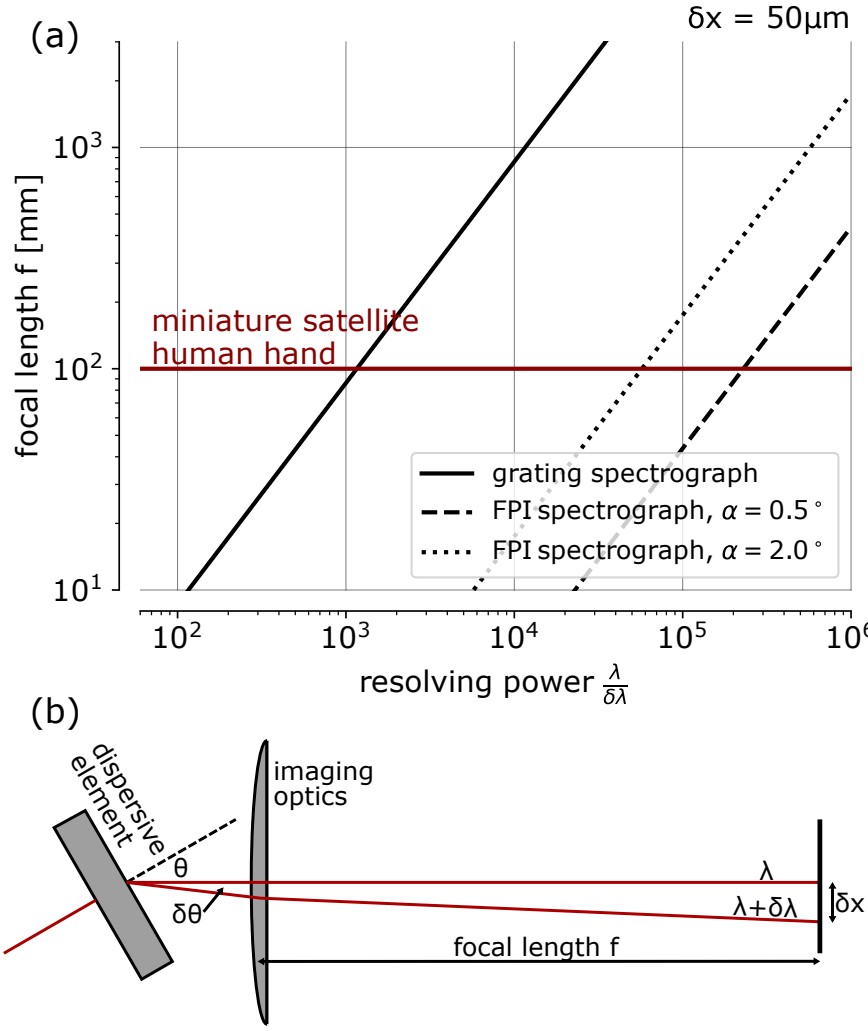

**Figure 3.** (a) Relationship between size (represented by the focal length) and resolving power of GS and FPI spectrograph for an ILF spatial dimension $\delta x = 50\,\mu m$. The size of a human hand or a miniature satellite is about $10\,cm$ (red line), determining favourable resolving power of the respective GS or FPI spectrograph for many applications. (b) Schematic of a spectrograph illustrating its fundamental aspects that determine the values in (a). The focal length $f$ mainly determines the overall spectrograph size.

the corresponding GS is about 1000 and thereby quite close to that used by moderate resolution DOAS measurements. The resolving power of the corresponding ($f = 10\,cm$) FPI spectrograph is in the range of $10^5$ and thereby capable of resolving

individual rovibronic absorption lines of trace gases in the UV and visible spectral range.

These considerations point towards the advantages of FPIs for high resolution spectroscopy, where they have been widely in use since more than a century (see Vaughan, 1989). Moreover, we illustrated that the fundamental differences of grating and FPI constitute different instrument sizes (or levels of mobility) for a given resolving power. However, these consideration do





not yet include the spectrograph's light throughput and hence the maximum achievable SNR, which is also decisive for most
atmospheric remote sensing applications.

## 3.2 Light throughput

In the following we derive the general relationship between the sensitivity of a spectrosopic measurement and the light through-
put of a spectroscopic instrument. The light throughput $k_{\mathrm{H}}$ of a spectrograph defines the conversion of incoming spectral radi-
ance $I$ (in units of $[\mathrm{photons\,s^{-1}\,mm^{-2}\,sr^{-1}\,nm^{-1}}]$) to a flux $J_{\mathrm{ph,H}}$ of photons with energies (or wavelengths) from within a
single spectral channel of the spectrograph (see Eq. 16, below).

The upper limit for the SNR of an atmospheric remote sensing measurement is often determined by photo electron shot noise,
i.e. by the number $N_{\mathrm{ph}} = J_{\mathrm{ph,H}} \cdot \delta t$ of photons detected within an exposure time period $\delta t$ (defining the measurement interval).
The noise of such a spectrum is given by $\sqrt{N_{\mathrm{ph}}}$ and thus, the photon SNR $\Theta$ of a spectrum can be approximated by:

$$\Theta \approx \frac{N_{\mathrm{ph}}}{\sqrt{N_{\mathrm{ph}}}} = \sqrt{I\,k_{\mathrm{H}}(\delta\lambda)\,\delta t} \tag{13}$$

This can be translated to the corresponding limits $\Delta S$ for the detection of trace gas column densities using the effective
differential absorption cross sections $\bar{\sigma}(\delta\lambda)$, which in many cases is a function of spectral resolution (compare Fig. 1):

$$\Delta S \approx \frac{1}{\bar{\sigma}(\delta\lambda)\,\Theta} = \frac{1}{\bar{\sigma}(\delta\lambda)\,\sqrt{I\,k_{\mathrm{H}}(\delta\lambda)\,\delta t}} \tag{14}$$

Here the crucial role of the light throughput of the instrument becomes obvious, especially when the radiance of the light source
(e.g. scattered sunlight) and the exposure time (e.g. time constant of the process to be studied) are fixed. Also, the choice of
$\delta\lambda$ is a compromise between optimal sensitivity (i.e. $\bar{\sigma}$, typically decreasing for increasing $\delta\lambda$) and optimal light throughput
(typically increasing for increasing $\delta\lambda$, see below). Particularly for trace gases with absorption cross sections consisting of
discrete lines (e.g. OH, water vapour or $O_2$), the sensitivity increases almost linearly with the spectral resolution as long as it
is much lower than the line width.

When broadband light sources are used, a linear dependency of the light throughput on $\delta\lambda$ is introduced. For line emitters,
where the sepctral width of the emitted line is smaller than $\delta\lambda$ (e.g. atomic emission lines) this is not the case (compare e.g.
Jacquinot, 1954). We here regard light sources that are broadband compared to $\delta\lambda$ (scattered or direct sunlight or incoherent
artificial light sources) and therefore include the factor $\delta\lambda$ in the light throughput quantification. Further, the light throughput
depends on the geometric beam acceptance of the optics (i.e. its étendue $E_{\mathrm{H}}$), which often introduces a further $\delta\lambda$ dependency
(see the following subsections). The spectrograph's étendue for a given spectral channel is approximated by the product of
surface area $A_{\mathrm{H}}$ and solid angle $\Omega_{\mathrm{H}}$ of the corresponding light beam:

$$E_{\mathrm{H}} \approx A_{\mathrm{H}} \cdot \Omega_{\mathrm{H}} \tag{15}$$

Losses at the optical components are accounted for by a factor $\mu$. From these effects, the light throughput can then be calculated
via:

$$k_{\mathrm{H}} = \frac{J_{\mathrm{ph,H}}}{I} = \mu\,\delta\lambda\,E_{\mathrm{H}}(\delta\lambda) \tag{16}$$


In the following, we compare the light throughput of FPI spectrographs with that of GSs for a given spectral resolution $\delta\lambda$. The losses at the optical components depend on their number, type and quality. We assume that $\mu$ (accounting for these losses) is always optimised and that, apart from the OSM (introducing about a factor of two), there is no substantial difference in $\mu$ for FPI spectrograph and GS. Thus, the light throughput is essentially determined by the étendue $E_{\mathrm{H}}$ of an individual spectral channel.

We derive the étendue $E_{\mathrm{H}}$ of GS and FPI spectrograph by approximating the surface area on the focal plane detector that is illuminated with light of a single spectral channel. The spectrographs imaging optics determines the corresponding beam solid angle.

Imaging magnification does not affect the étendue (which is one of the reasons why the étendue is a universal measure of a spectrograph's quality) since it only converts solid angle into surface area and vice versa. Therefore, for light throughput
comparison we can ignore magnification and always assume ideal 1:1 imaging (i.e. collimating and focusing optics with the same focal length).

Investigating the light throughput per wavelength interval $\delta\lambda$ allows the comparison of spectrograph setups with respect to their photon shot noise limited SNR.

### 3.2.1   Étendue of a grating spectrograph

For a simple GS, as typically used for DOAS measurements, the above definition of $E_{\mathrm{H}}$ might seem a bit artificial, since the étendue per spectral channel $\delta\lambda_{\mathrm{GS}}$ equals the étendue of the entrance optics. Assuming ideal 1:1 imaging, the surface area $A_{\mathrm{H,GS}}$ on the detector that is illuminated by light from within $\delta\lambda_{\mathrm{GS}}$ is determined by the illuminated slit area, i.e. by the illuminated slit height $h_{\mathrm{S}}$ and width $w_{\mathrm{S}}$. The slit width determines the spectral resolution via the GS's linear dispersion $D_{\mathrm{GS}} := \frac{\mathrm{d}x}{\mathrm{d}\lambda}$ along the dispersion direction $x$. Because of the 1:1 imaging, $A_{\mathrm{H,GS}}$ at the detector is given by:

$$A_{\mathrm{H,GS}} = w_{\mathrm{S}}\,h_{\mathrm{S}} = \delta\lambda_{\mathrm{GS}}\,D_{\mathrm{GS}}\,h_{\mathrm{S}} \tag{17}$$

The corresponding imaging beam solid angle $\Omega_{\mathrm{H}}$ can be calculated from the F-number $F_{\mathrm{GS}} = \frac{f}{b}$ of the GS's imaging optics, according to the approximation for higher F-numbers:

$$\Omega_{\mathrm{H,GS}} \approx \frac{\pi}{4\,F_{\mathrm{GS}}^2} = \frac{\pi\,b^2}{4\,f^2} \tag{18}$$

with the imaging optics' (or the grating's) circular CA $b$ and its focal length $f$. The étendue of a GS is then:

$$E_{\mathrm{H,GS}} \approx A_{\mathrm{H,GS}} \cdot \Omega_{\mathrm{H,GS}} \approx \frac{\pi}{4\,F_{\mathrm{GS}}^2}\,w_{\mathrm{S}}\,h_{\mathrm{S}} = \frac{\pi}{4\,F_{\mathrm{GS}}^2}\,\delta\lambda_{\mathrm{GS}}\,D_{\mathrm{GS}}\,h_{\mathrm{S}} \tag{19}$$

 In the spectral ranges regarded in this study, due to the availability of appropriate gratings, the GS resolving power is basically determined by slit imaging. When the grating is optimised to the operating wavelength (i.e. $r_{\mathrm{g}} \approx m\,\lambda$, see above, or $\kappa\,r_{\mathrm{g}} = m\,\lambda$ with $\kappa \approx 1$), the GS resolving power is determined by slit width and focal length (see Appendix B for details):

$$\frac{\lambda}{\delta\lambda_{\mathrm{GS}}} = \kappa\,\frac{f}{w_{\mathrm{S}}} \tag{20}$$



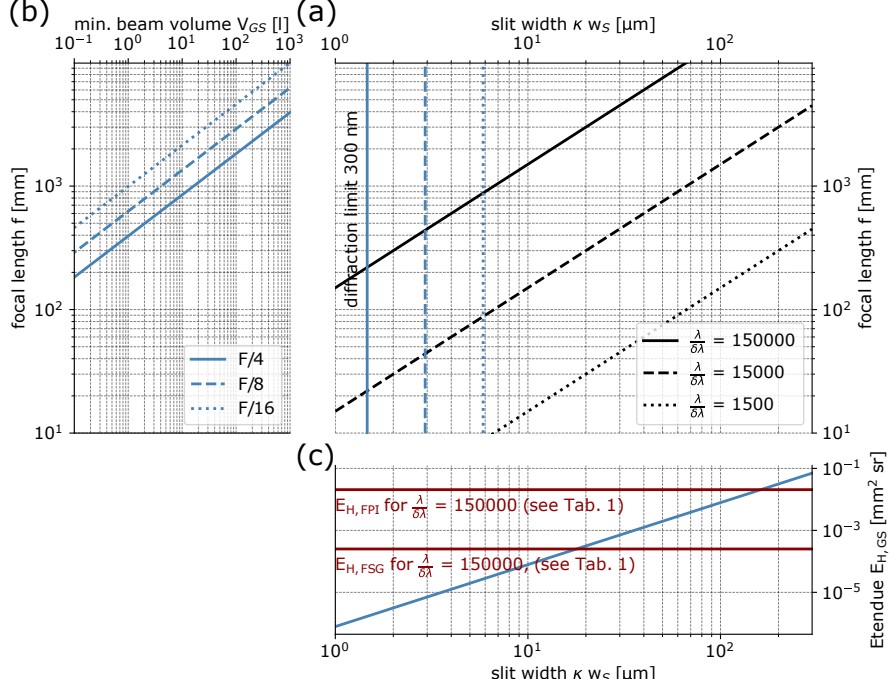

**Figure 4.** Combined visualisation of Eqs. (20) - (22) and (24). For three exemplary resolving powers (1500, 15000, 150000), the possible slit width to focal length ratios are shown (a). The focal length determines the spectrograph's size scaling (b), while the slit width determines its étendue (blue line in c). In red the étendue of the FPI spectrograph with grating OSM (for an incidence angle of $0.5°$) and the total étendue of the FPI (i.e. the FPI spectrograph with interferometric OSM) with the specifications given in Tab. 1 are shown.

Without exact knowledge of the factor $\kappa$ (which is around unity and accounts for slight inaccuracies in the assumptions made) this relation allows to evaluate how the size and the étendue of a particular GS change with its slit width and focal length for constant resolving power (see Fig. 4). As a measure for the spectrograph's size scaling a minimum "beam volume" $V_{\mathrm{GS}}$ is determined by the light cone constrained by the F-number and the focal length:

$$V_{\mathrm{GS}} = \frac{1}{12}\,\pi\,f\,b^2 = \frac{1}{12}\,\pi\,\frac{f^3}{F_{\mathrm{GS}}^2} \tag{21}$$

While representing the lowest boundary for the absolute size of the spectrograph's optical setup it describes the scaling of a GS's volume and mass with the third power of its focal length for a constant F-number (see also Platt et al., 2021).

The resolving power of such an idealised GS can now be increased by either increasing the focal length or by narrowing the entrance slit (see Fig. 4). Increasing the focal length leads to a larger and heavier instrument and is therefore limited by mobility requirements. Narrowing the entrance slit reduces the étendue of the GS. The theoretical lower bound is given by diffraction at 345 the entrance slit, i.e.:

$$w_{\mathrm{S,min}} \approx 1.22\,F_{\mathrm{GS}}\,\lambda \tag{22}$$



In practice, imaging aberrations limit the resolving power for narrow slit widths. In particular, aberrations will limit the slit height of the GS, which substantially influences the GS étendue (see Eq. (4)). Approximating the maximum possible slit height based on an empirical quantification of the astigmatism of GSs by Fastie (1952) leads to the simple expression (see Appendix C):

$$h_{\mathrm{S}} \approx w_{\mathrm{S}} F_{\mathrm{GS}}^2 \qquad (23)$$

By inserting this relationship into Eq. (19) the expression for the GS étendue is further simplified to:

$$E_{\mathrm{H,GS}} \approx \frac{\pi}{4} w_{\mathrm{S}}^2 \propto \delta\lambda_{\mathrm{GS}}^2 \qquad (24)$$

Surprisingly, the F-number cancels, which is because small F-numbers increase the accepted beam solid angle of the GS, while, at the same time and by the same amount, reducing the allowed slit height through imaging aberrations. In principle, this introduces a dependence of the GS étendue to the square of $\delta\lambda_{\mathrm{GS}}$, which further stresses the problems of high resolution GS. This does not mean that the F-number can be chosen arbitrarily. To avoid further distortions, the slit height must remain much smaller than the CA of the imaging optics.

Correcting aberrations (like the astigmatism) is possible but onerous. Large imaging spectrographs can reach large slit heights with a low F-number for instance by using lens optics to avoid off axis imaging and thereby largely reducing aberration (see e.g. Crisp et al., 2017). This will not be considered in this study, where we focus on mobile spectrographs.

### 3.2.2 Etendue of the FPI spectrographs

In order to assess the étendue of the FPI spectrographs it is useful to first regard the étendue of a single FPI order, ignoring for the moment the influence of the OSM. For instance an idealized bandpass filter or a FSR much larger than the spectral band of the light source could be assumed. By assessing the transmission solid angles $\Omega_{\mathrm{H,FPI}}$ of a FPI order (see Appendix D) we find the étendue of the FPI, which is, for a given resolving power, only dependent on the FPI CA $b_{\mathrm{FPI}}$:

$$E_{\mathrm{H,FPI}} \approx \frac{\pi^2}{2} b_{\mathrm{FPI}}^2 \frac{\delta\lambda_{\mathrm{FPI}}}{\lambda} \qquad (25)$$

Consequently, in the focal plane of a lens that is placed behind the FPI (lens 2 in Fig. 2a), the appearing rings corresponding to a wavelength interval $\delta\lambda_{\mathrm{FPI}}$ (see Fig. 2c and 2d) have the same surface area and the étendue of all spectral channels is the same. For a given FPI CA and resolving power, $E_{\mathrm{H,FPI}}$ (as given by Eq. 25) states an upper limit for the étendue of a FPI spectrograph.

For a FPI spectrograph with interferometric OSM this étendue can be reached if the étendue of all the OSM components are equal or larger than $E_{\mathrm{H,FPI}}$. This should not be a problem, since the interferometric OSM components are FPIs or interference filters with similar or lower resolving powers and therefore higher étendue for the same CA (see Appendix F for details).

For the grating OSM things are a bit more complicated. We assume an order sorting GS (OSGS) with a spectral resolution of about the FPI's FSR (i.e. $R_{\mathrm{FPI}} = R_{\mathrm{OSGS}} \cdot \mathcal{F}$). The spectrum of the OSGS can for instance be re-imaged by the FPI imaging optics (Fig. 2a). Thereby the radially symmetric FPI spectral transmission overlaps with the OSGS spectrum, resulting in stripes





(along the OSGS slit dimension) that isolate individual FPI transmission orders. Particularly, this will introduce a FPI incidence angle dependence to the étendue. The étendue of the FPI spectrograph with grating OSM $E_{\mathrm{H,FSG}}$ can be approximated by (see

Appendix E):

$$E_{\mathrm{H,FSG}} \approx \frac{\pi}{4\,F^2}\,w_{\mathrm{S}}\,\frac{f_2}{\alpha}\,\frac{\delta\lambda_{\mathrm{FPI}}}{\lambda} \approx \frac{w_{\mathrm{S}}}{2\,\pi\,f_2\,\alpha}\,E_{\mathrm{H,FPI}} \tag{26}$$

As expected the étendue equals the étendue of the OSGS with the slit height replaced by the radial extent of an FPI transmission ring with the spectral width of $\delta\lambda_{\mathrm{FPI}}$. Further, it can be expressed as a fraction of the total étendue (Eq. (25)) of the used FPI. The expression approximates only a part of the total spectrum recorded with such a FPI spectrograph (i.e. where grating

dispersion and FPI dispersion are approximately perpendicular). It is, however, representative for large parts of the spectrum. The OSGS resolution $\delta\lambda_{\mathrm{OSGS}}$ needs to approximately equal the FSR $\Delta\lambda_{\mathrm{FPI}}$ of the FPI. This means that the slit width $w_{\mathrm{S}}$ of the OSGS (and thereby $E_{\mathrm{H,FSG}}$) can be increased if the FSR of the FPI is increased. In order to keep the spectral resolution $\delta\lambda_{\mathrm{FPI}}$ constant, the same increase is required for the finesse. For increasing slit width, FSR and finesse, $E_{\mathrm{H,FSG}}$ converges to $E_{\mathrm{H,FPI}}$. Since then less FPI orders are sampled, the total wavelength coverage decreases. This allows for instance to adjust the

spectral coverage and the étendue according to a specific application.

For Eq. (26) to hold, the F-numbers of OSGS and FPI imaging optics need to be matched. The focal length $f_2$ is thus determined by the FPI's CA and the OSGS's F-number. Figure 2c and 2d illustrate the étendue differences of interferometric and grating OSM.

### 3.2.3   Comparison of FPI spectrographs and GS

With the above evaluation of the étendue we can compare the light throughput and SNR of FPI spectrographs with a GS for a given resolving power. Further we can relate the results to moderate resolution GSs with known absolute SNR. This allows approximating the absolute SNR of high resolution FPI spectrographs for atmospheric remote sensing applications. Table 1 summarises the results.

In order to reach spectral resolutions of the order of single rotational trace gas absorption lines a resolving power of 150000

is assumed, which corresponds to a $2\,\mathrm{pm}$ spectral resolution at $300\,\mathrm{nm}$. A $100\,\mathrm{mm}$ focal length facilitates the mobility of the spectrograph (Sect. 3.1). As found in Sect. 3.2.1 (compare Fig. 4) the high resolution GS can not be implemented with $100\,\mathrm{mm}$ focal length (due to diffraction at the entrance slit) and therefore uses optics with a focal length of $1\,\mathrm{m}$. We also assume the same F-number of $F = 4$ for all spectrographs. These assumptions mainly determine the étendue of the spectrographs.

We assume here that for the FPI spectrograph with interferometric OSM, the element with the highest resolving power (i.e.

the FPI with $R = 150000$) limits the étendue (see Eq. (25) and, for further details, Appendix F). The FPI spectrograph with grating OSM requires the FSR of the FPI to be matched with the OSGS spectral resolution. We assume a FPI with a finesse of 100 and therefore need a OSGS with $R = 1500$. A finesse of 100 for the given FPI dimensions is challenging but possible to manufacture for the UV. For larger wavelengths even higher finesses (i.e. higher spectrograph light throughputs) can be reached. The étendue of the FPI spectrograph with grating OSM was calculated for a representative FPI incidence angle of

$\alpha = 0.5°$. For the light throughput comparison the OSMs are accounted for by a loss factor of $0.5$.





**Table 1.** Comparison of FPI spectrograph and GS. All spectrographs have a F-number of 4 and to ensure mobility a focal length of $100\,\mathrm{mm}$, except for the high resolution GS (see Sect. 3.2.1 for details). The light throughput and SNR are calculated relative to that of a moderate resolution GS, commonly used for DOAS measurements, and thus with known SNR.

| | | | FPI spectrograph | | | grating spectrograph | | |
| | | | interferom. OSM | grating OSM | | high res. | moderate res. | |
| | | | | | OSGS | | optimised | typ. DOAS |
| quantity | symbol | unit | | | $\alpha = 0.5°$ | | | |
| --- | --- | --- | --- | --- | --- | --- | --- | --- |
| resolving power | $R$ | | 150000 | 150000 | 1500 | 150000 | 600 | ca. 600 |
| spect. res. @ 300 nm | $\delta\lambda$ | nm | 0.002 | 0.002 | 0.2 | 0.002 | 0.5 | ca. 0.5 |
| principal focal length | $f$ | mm | 100 | 100 | 100 | 1000 | 100 | 75 |
| F-number | $F$ | | 4 | 4 | 4 | 4 | 4 | 4 |
| slit width | $w_S$ | µm | - | - | 67 | 6.7 | 167 | 100 |
| slit height | $h_S$ | µm | - | - | 268 | 26.8 | 668 | 400 |
| grating/FPI CA | $b$ | mm | 25 | 25 | 25 | 250 | 25 | 18.75 |
| étendue | $E_H$ | mm$^2$ sr | $2.06\cdot10^{-2}$ | $2.51\cdot10^{-4}$ | | $3.54\cdot10^{-5}$ | $2.19\cdot10^{-2}$ | $1.96\cdot10^{-3}$ |
| rel. loss | $\frac{\mu}{\mu_0}$ | | 0.5 | 0.5 | | 1 | 1 | 1 |
| rel. light throughput | $\frac{k_H}{k_{H,0}}$ | | $1.05\cdot10^{-2}$ | $1.28\cdot10^{-4}$ | | $3.61\cdot10^{-5}$ | 11.15 | 1 |
| rel. SNR | $\frac{\Theta}{\Theta_0}$ | | $1.02\cdot10^{-1}$ | $1.13\cdot10^{-2}$ | | $6.00\cdot10^{-3}$ | 3.34 | 1 |
| rel. volume and mass | $\frac{V}{V_0}$ | | 1-2 | 1-2 | | 1000 | 1 | 1 |
| rel. resolving power $\sqrt{\text{light throughput}}$ product per instrument volume | $\frac{R V_0}{V}\sqrt{\frac{k_H}{k_{H,0}}}$ | | $7685 - 15370$ | $849 - 1697$ | | 1 | 2003 | 600 |

In practice, a moderate resolution DOAS GS with $f = 75\,\mathrm{mm}$ typically has a resolving power of 600 (i.e. a spectral resolution of 0.5 nm at 300 nm) and a $100\,\mathrm{\mu m}$ wide slit is used together with for instance an optical fibre of $400\,\mathrm{\mu m}$ determining the illuminated slit height (see e.g. Platt and Stutz, 2008). Such setups are able to record spectra of scattered sky light with SNRs of several thousands in the UV spectral range within about one minute integration time (see e.g. Lauster et al., 2021). In addition, we determined the light throughput of an (with respect to our formalism) optimised GS with the same moderate resolving power and 100 mm focal length. Its light throughput is by about an order of magnitude higher than that of moderate resolution GS presently in use.

Compared to such compact moderate resolution GSs that are in use for DOAS measurements, the FPI spectrograph with interferometric OSM exhibits a by a factor of 100 lower light throughput with a 250 times higher spectral resolution. Consequently, for a given integration time the photon SNR of the high resolution spectrum of the FPI spectrograph is only about 10 times





lower than that of a compact moderate resolution GS. For the spectrum of a grating OSM FPI spectrograph the corresponding SNR is 100 times lower for the same gain in spectral resolution. However, a considerably larger wavelength range is covered compared to the interferometric OSM version.

The high resolution GS, despite of its volume that is already about 1000 times the volume of the other spectrographs, yields

even only about half the SNR of the grating OSM FPI spectrograph.

Extending the FPI's CA to $250\,\mathrm{mm}$ would yield the 250 fold increase of spectral resolution with the same SNR as a compact moderate resolution DOAS spectrograph. If such an FPI could be manufactured the corresponding spectrograph would have a focal length of about $1\,\mathrm{m}$. The corresponding high resolution GS with the same SNR would need a focal length of about $15\,\mathrm{m}$.

## 4   Implications for atmospheric remote sensing and prototype

### 4.1   Implications for atmospheric remote sensing

FPI spectrographs offer a way to reach large resolving powers with, compared to GSs, a largely reduced impact on the SNR and maintaining a mobile instrument setup. This might allow substantially lower detection limits of trace gas measurements in the near UV to near IR spectral range or to increase their spatial or temporal resolution.

When regarding noise limited trace gas detection limits (as introduced in Eq. 14), we find that for many gases in the near UV to

near IR the effective differential absorption cross section (and thus the sensitivity of the measurement) increases with spectral resolution. For absorbers with discrete lines, e.g. OH, water vapour, or $O_2$, the sensitivity increase will be almost linear to the increase in spectral resolution (i.e. for our example a factor of ca. 250). For such gases, this effect outweights the effect of reduced light throughput (0.01 compared to moderate resolution GSs, Tab. 1) and the corresponding noise limited detection limits of the FPI spectrograph with interferometeric OSM will be by a factor of $(250 \cdot \sqrt{0.01})^{-1} = 0.04$ (0.4 for grating OSM)

lower than that of common DOAS measurements with moderate spectral resolution. By reducing the temporal resolution of FPI spectrograph measurements by a factor of 100 (i.e. increasing the exposure time e.g. from $30\,\mathrm{s}$ to $50\,\mathrm{min}$) the same photon SNR as that of moderate resolution DOAS measurements can be reached, reducing the detection limits by another order of magnitude.

In addition, the increase of sensitivity comes with a massive increase of selectivity, since the high spectral resolution on one

hand allows using much more specific absorption structures for gas detection and, on the other hand, reduces or removes the influence of undersampled Fraunhofer lines for sun light measurements. Thereby, detection limits can further be significantly lowered with respect to moderate resolution measurements, which are in many cases also limited by cross interferences (see e.g. Vogel et al., 2013). Consequently, also the feasibility of distinguishing trace gas isotopes is strongly enhanced and line broadening effects could add valuable information to retrievals of vertical atmospheric trace gas distributions.

Similar advantages are expected e.g. for the passive quantification of solar induced fluorescence of chlorophyll by in-filling of narrow solar Fraunhofer lines with increased spectral resolution (see e.g. Plascyk and Gabriel, 1975; Grossmann et al., 2018). The following simple example outlines the impact FPI spectrographs might have for atmospheric sciences: According to the above assessment, a high resolution FPI spectrograph records a spectrum with a SNR $\Theta$ of 3333 in about one hour integration





time. For scattered sunlight measurements in the UV, the tropospheric light path can reach about $10\,\mathrm{km}$. The absorption cross
section of OH $\bar{\sigma}_{\mathrm{OH}}$ at around $308\,\mathrm{nm}$ reaches about $1.5 \cdot 10^{-16}\,\mathrm{cm^2\,molec^{-1}}$ (see Rothman et al., 2013). The detection limit
of OH concentrations $\Delta c_{\mathrm{OH}}$ (see Eq. (14)) would then be

$$\Delta c_{\mathrm{OH}} \approx \frac{\Delta S_{\mathrm{OH}}}{L} = \frac{1}{\Theta\,\bar{\sigma}_{\mathrm{OH}}\,L} = 2 \cdot 10^6\,\mathrm{molec\,cm^{-3}} \tag{27}$$

This is already in the range of tropospheric background OH concentrations (see e.g. Stone et al., 2012). This detection limit
can further be lowered by using active light sources like LEDs or Xe lamps instead of scattered sunlight or by the use of larger
FPIs or arrays of parallel FPI spectrographs.

Further, as assessed in Sect. 1.4, FPI spectrographs are expected to have similar advantages over FTS and GS measurements in
the near IR. Thus, FPI spectrographs could also substantially improve remote sensing measurements of for instance greenhouse
gases (e.g. $CO_2$ or $CH_4$) or CO in Earth's atmosphere. Instead of the large spectral coverage with high resolution reached by
FTS, several FPI spectrographs could record spectra in different spectral windows that are relevant for the trace gas retrieval
(e.g. an additional spectral window for $O_2$ light path information, see e.g. Crisp et al., 2017).

An important aspect, regarding the named and quantified benefits of FPI spectrographs, is that the low level of complexity and
the high mobility of presently used moderate resolution GS measurements is maintained.

## 4.2 FPI spectrograph prototype

As a proof of concept we built a prototype of a FPI spectrograph with grating OSM at the Institute of Environmental Physics in
Heidelberg (see Fig. 5a). It operates at around $308\,\mathrm{nm}$. A FPI with high finesse (ca. 95) across a CA of $5\,\mathrm{mm}$ and a resolving
power of ca. 148000 (supplied by SLS Optics Ltd.) was used together with a compact OSGS. We recorded a spectrum of
light of a UV LED that traversed a burner flame (see Fig. 5a) containing large amounts of OH (typically several thousands of
ppm, see e.g. Cattolica et al., 1982). For a light path of about $1\,\mathrm{cm}$ this leads to optical densities >1 for many OH lines (see
OH absorption spectrum in Fig. 1, which is slightly altered due to the high temperature, see Rothman et al., 2013). Figure
5b shows the corresponding spectrum recorded by the FPI spectrograph prototype. The bright vertical stripes originate from
a slight overlap of the individual FPI orders and thus also indicate their boundaries (compare Fig. 2b and 2c). The dark spots
correspond to individual OH absorption lines. This is verified by calculating the intensity distribution using an instrument
model and literature OH absorption data. The orange box in Fig. 5b shows the region of the spectrum that is modeled in Fig.
5c. The locations of the individual OH absorption lines (dark spots) are clearly reproduced by the model, confirming the high
resolving power.

Compared to the FPI spectrograph we assumed in Sect. 3.2.3 the light throughput of this prototype instrument is reduced
according to its smaller CA (i.e. by a factor of about 25, see Eq. (25)). The mobility advantages of FPI spectrographs as derived
in Sect. 3.1 are already demonstrated by this still rudimentary prototype. Its volume is below 8 litres and it weighs less than
$5\,\mathrm{kg}$. The FPI can be replaced by an FPI with larger CA without significantly impacting the instrument size.





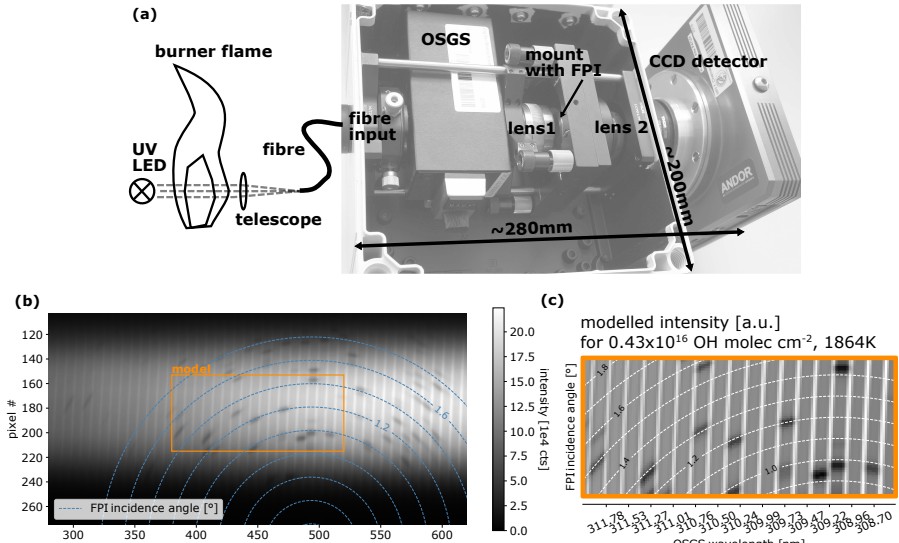

**Figure 5.** Prototype of a FPI spectrograph with grating OSM recording an absorption spectrum of OH in a burner flame: (a) instrument and experiment setup: Light from a UV LED traverses a burner flame (containing a high amount of hot OH) before being directed to the FPI spectrograph via a telescope and a fibre. The FPI imaging optics reimages the moderate resolution spectrum of the OSGS (compare Fig. 2). (b) Recorded spectrum image: The vertical bright stripes arise from slight overlapping of FPI orders. Dark spots indicate the individual OH absorption lines. The dashed blue lines indicate rings of equal FPI incidence angle $\alpha$. (c) Modeled intensities (using high temperature OH absorption data from Rothman et al., 2013) for a part of the measured spectrum (orange box) with an instrument model show excellent agreement.

A comprehensive description of this and further prototype instruments and the instrument models would go beyond the scope of this work and will be treated in further publications.

# 5 Conclusions

We compared the performance of high resolution spectrographs using gratings or FPIs. Increasing the spectral resolution of a
GS results in the loss of its mobility and light throughput advantages, and thereby its applicability to many atmospheric studies. In contrast, the implementation of mobile FPI spectrographs with high resolving powers is possible (as shown by the presented prototype) and can further yield a much larger light throughput than a GS with the same (high) resolving power. Compared to moderate resolution GSs (as used in conventional DOAS measurements), FPI spectrographs with nowadays available optical components and a 250 fold spectral resolution (e.g. 2 pm instead of 0.5 nm at 300 nm) yield a light throughput that is only
by a factor of 100 smaller for an instrument of the same size. In contrast to that, the corresponding high resolution GS, which can only be implemented with about the 1000 fold volume, yields only about $4 \cdot 10^{-5}$ of the moderate resolution GS's light throughput.





Similarly to the resolving power luminosity product used by e.g. Jacquinot (1954) to generally compare FPIs to gratings we can define a figure to quantify the applicability of spectroscopic instruments to atmospheric remote sensing studies with enhanced mobility requirements (such as for instance measurements in remote areas or satellite instruments). This would then be the product of resolving power and the square root of the light throughput (proportional to the inverse trace gas detection limits) per instrument volume. For a resolving power of 150000 this figure is at least 3 - 4 orders of magnitude larger for FPI spectrographs compared to the GS.

The employment of mobile high resolution FPI spectrographs would on the one hand substantially increase the SNR of high resolution measurements in the atmosphere and at the same time substantially increase the mobility of measurement instrumentation. This comes basically at the cost of spectral coverage of the spectrograph, which, however, for many applications should not be a problem.

The impact on atmospheric remote sensing measurements may be outlined by the following examples:

(1) More trace gases (like e.g. tropospheric OH) could be detectable by relatively simple passive or active absorption measurements.

(2) In many cases, detection limits of trace gases (e.g. $SO_2$, $H_2O$, HCHO, ClO, BrO) routinely quantified by moderate spectral resolution DOAS measurements could be significantly lowered through the enhancement of sensitivity and selectivity through high spectral resolution.

(3) Alternatively, the temporal or spatial resolution of such measurements could be enhanced.

(4) From passive measurements using sunlight, absolute (rather than differential) column density measurements of trace gases absorbing in the UV and visible wavelength range could become possible (e.g. evaluation between Fraunhofer lines).

(5) Through the increase in spectral resolution, the capability of separating trace gas isotope absorption is enhanced.

(6) Line broadening could be quantified to add valuable information to the retrievals of vertical trace gas distributions.

(7) Radiative transfer in haze or clouds can be studied with high resolution measurements of $O_2$ rotational lines.

(8) Increased spectral resolution also enhances the sensitivity of chlorophyll fluorescence quantification through in-filling of Fraunhofer lines and similar studies.

(9) FPI spectrographs are expected to similarly improve trace gas measurements in the near IR as presently performed with FTS (e.g. quantification of green house gases in the atmosphere).

All in all, the results of this study suggest that high resolution spectroscopy with mobile FPI spectrographs has the potential to substantially advance atmospheric trace gas remote sensing, thereby opening the door to many new insights into processes in Earth's atmosphere.

## Appendix A: List of abbreviations and symbols

See Tabs. A1 and A2





**Table A1.** List of abbreviations

| | |
|---|---|
| CA | clear aperture |
| DOAS | differential optical absorption spectroscopy |
| FPI | Fabry-Pérot interferometer |
| FSG | FPI spectrograph with grating OSM |
| FSR | free spectral range |
| FTS | Fourier transform spectroscopy |
| FWHM | full width at half maximum |
| GS | grating spectrograph |
| ILF | instrument line function |
| IR | infrared |
| OSGS | order sorting grating spectrograph |
| OSM | order sorting mechanism |
| SNR | signal to noise ratio |
| UV | ultraviolet |

**Appendix B: The GS resolving power is mainly limited by slit imaging**

The resolving power of the grating is limited by the number of illuminated grating rules $N_\mathrm{g}$ (i.e. $\frac{\lambda}{\delta\lambda} = N_\mathrm{g}$). This requires $\frac{f}{F_\mathrm{GS}\, r_\mathrm{g}}$ to be larger than the intended resolving power, which is almost always fulfilled by commonly used GSs implementations. For an ideal choice of the grating, its effective ruling distance $r_\mathrm{eff} = r_\mathrm{g}\cos\theta_\mathrm{g}\, m^{-1}$ (see Eqs. (7) and (8)) should be in the range of the measured wavelength. Gratings with that specification are available for all spectral ranges of interest for this study. Thus, one can conclude that the GS resolving power is in most cases limited by slit imaging. We assume here that $r_\mathrm{eff} = \frac{\lambda}{\kappa}$, with $\kappa$

being close to unity and accounting for any uncertainties in the assumptions. With the linear dispersion $D_\mathrm{GS} = f\, r_\mathrm{eff}^{-1}$, we then find the relation:

$$w_\mathrm{S} = \delta\lambda_\mathrm{GS}\, D_\mathrm{GS} = \frac{\delta\lambda_\mathrm{GS}}{r_\mathrm{eff}}\, f = \frac{\delta\lambda_\mathrm{GS}}{\lambda}\, \kappa\, f \;\Leftrightarrow\; \frac{\lambda}{\delta\lambda_\mathrm{GS}} = \kappa\, \frac{f}{w_\mathrm{S}} \tag{B1}$$

**Appendix C: Aberration limited slit height of a compact GS**

We will approximate the maximum possible slit height based on an empirical quantification of the astigmatism of GSs by

Fastie (1952). The astigmatism is the deviation $\Delta f$ of focal length in along and across slit direction, introduced by off-axis imaging with e.g. spherical mirrors. It is found to be proportional to the focal length and to the square of the angular distance $\phi$ of the slit to the normal of the focussing/collimating mirror. The entrance slit and the focal plane of the GS are separated by at least the grating's diameter $b$, hence, the lower limit of $\phi$ is given by $\frac{b}{2f} = \frac{1}{2F_\mathrm{GS}}$. With that, Fastie's empirical astigmatism



**Table A2.** List of symbols

| | | | |
|---|---|---|---|
| $\lambda$ | wavelength | | |
| $\delta\lambda$ | spectral resolution, spectral ILF FWHM | $R$ | resolving power |
| $\Gamma$ | optical path difference of FPI | $d$ | FPI mirror separation |
| $n$ | refractive index of FPI medium | $\alpha$ | incidence angle of light onto FPI |
| $m$ | order of FPI fringe or grating dispersion | $\lambda_\mathrm{m}$ | wavelength at FPI fringe with order $m$ |
| $\Delta\lambda_\mathrm{FPI}$ | FSR of the FPI | $\mathcal{F}$ | finesse of the FPI |
| $H$ | ILF | $\Lambda$ | wavelength coverage |
| $B$ | diameter of circular entrance aperture | $f$ | focal length |
| $\theta$ | general dispersion deflection angle | $\delta\theta$ | small, linearised change in $\theta$ |
| $r_\mathrm{g}$ | ruling distance of a grating | $\delta x$ | spatial separation in the focal plane through $\delta\theta$ |
| $k_\mathrm{H}$ | light throughput per spectral channel | $I$ | radiance |
| $J_\mathrm{ph,H}$ | photon flux per spectral channel | $N_\mathrm{ph}$ | number of photons |
| $\Theta$ | SNR | $\delta t$ | measurement interval, exposure time |
| $\Delta S$ | detection limit for a trace gas (column density) | $\bar{\sigma}$ | effective absorption cross section of a trace gas |
| $E_\mathrm{H}$ | etendue per spectral channel | $\Omega_\mathrm{H}$ | beam solid angle per spectral channel |
| $A_\mathrm{H}$ | surface area of beam cross section per spectral channel | $\mu$ | factor accounting for losses at optical components |
| $w_\mathrm{S}$ | slit width | $h_\mathrm{S}$ | slit height |
| $D_\mathrm{GS}$ | linear dispersion of a GS | $F$ | F-number |
| $b$ | CA | $\kappa$ | uncertainty factor around unity |
| $V$ | minimum beam volume of a spectrograph | | |

quantification can be expressed by focal length and F-number of the GS:

$$\Delta f = 0.4\, f\phi^2 = 0.1\frac{f}{F_\mathrm{GS}^2} \tag{C1}$$


The spread $\Delta L$ of an imaged point within the slit area along the defocussed astigmatism direction on the GS focal plane is then:

$$\Delta L = \frac{\Delta f}{F_\mathrm{GS}} \tag{C2}$$

Since for a GS, sharp imaging is only important in dispersion direction, its optics is always focussed to the focal length in
across slit direction. The astigmatism spread is then directed in along slit direction and therefore negligible for the spectral imaging. However, due to the radial symmetry of the imaging mirrors, the across slit component of the astigmatism increases with the distance from the slit center (assuming the slit is centered at the imaging plane). For the ends of the slit this component is given by the ratio of slit height $h_\mathrm{S}$ and the separation of entrance slit and slit image, which at least equals the grating's CA $b$. This means that at the slit ends the slit image is widened by:

$$w_\mathrm{S,ast} = \Delta L \frac{h_\mathrm{S}}{b} = 0.1\frac{h_\mathrm{S}}{F_\mathrm{GS}^2} \tag{C3}$$






When allowing for a slit widening by a tenth of the width of the slit image we find the slit height to be limited to:

$$h_\mathrm{S} = w_\mathrm{S}\, F_\mathrm{GS}^2 \tag{C4}$$

## Appendix D: Étendue of a FPI

If the FPI CA $b_\mathrm{FPI}$ is illuminated with a divergent light beam only light with the wavelength between $\lambda_\mathrm{m}^0 = \frac{2\,dn}{m}$ ($\lambda_\mathrm{m}$ for $\alpha = 0$)

and $\lambda_\mathrm{m}^0 - \delta\lambda_\mathrm{FPI}$ will be transmitted in the central beam part (limited by the incidence angle inducing a spectral shift of the FPI spectrum by $\delta\lambda_\mathrm{FPI}$, see Fig. 2). Each wavelength interval corresponds to an incidence angle interval limiting the solid angle of the respective transmitted beam. With Eqs. (1) and (2) and a cosine aproximation the incidence angle $\alpha$ corresponding to the transmission peak wavelength $\lambda_\mathrm{m}$ is determined:

$$\cos\alpha = \frac{\lambda_\mathrm{m}\, m_{FPI}}{2\,dn} = \frac{\lambda_\mathrm{m}}{\lambda_\mathrm{m}^0} \approx 1 - \frac{\alpha^2}{2} \Leftrightarrow \alpha \approx \sqrt{2\left(1 - \frac{\lambda_\mathrm{m}}{\lambda_\mathrm{m}^0}\right)} \tag{D1}$$

and thus for $\lambda_\mathrm{m} = \lambda_\mathrm{m}^0 - \epsilon$:

$$\alpha(\epsilon) \approx \sqrt{2\frac{\epsilon}{\lambda_\mathrm{m}^0}} \tag{D2}$$

where $\epsilon$ denotes the spectral displacement of $\lambda_\mathrm{m}$ with respect to $\lambda_\mathrm{m}^0$ (see Fig. 2b). The solid angle $\Omega_\mathrm{H,FPI}$ of a transmitted light beam with wavelength between $\lambda_\mathrm{m}^0 - p\,\delta\lambda_\mathrm{FPI}$ and $\lambda_\mathrm{m}^0 - (p+1)\,\delta\lambda_\mathrm{FPI}$ ($p$ being a positive real number) is then approximated by:

$$\Omega_\mathrm{H,FPI} \approx \pi(\alpha((p+1)\,\delta\lambda_\mathrm{FPI})^2 - \alpha(p\,\delta\lambda_\mathrm{FPI})^2) = 2\pi\frac{\delta\lambda_\mathrm{FPI}}{\lambda_\mathrm{m}^0} \tag{D3}$$

This means that the transmission solid angle of a FPI for a wavelength interval $\delta\lambda_\mathrm{FPI}$ is independent of the incidence angle and the étendue $E_\mathrm{H,FPI}$ for a beam with wavelength within $\delta\lambda_\mathrm{FPI}$ traversing the FPI CA ($A_\mathrm{H,FPI} = \frac{\pi}{4}\,b_\mathrm{FPI}^2$) is:

$$E_\mathrm{H,FPI} \approx \frac{\pi^2}{2}\, b_\mathrm{FPI}^2\, \frac{\delta\lambda_\mathrm{FPI}}{\lambda_\mathrm{m}^0} \approx \frac{\pi^2}{2}\, b_\mathrm{FPI}^2\, \frac{\delta\lambda_\mathrm{FPI}}{\lambda} \tag{D4}$$

## Appendix E: Étendue of a FPI spectrograph with grating OSM

We can assume that the focal plane of an GS (i.e. its spectrum) is re-imaged with a FPI imaging optics (as that shown in Fig. 2a) with a matched F-number. Further, the spectral resolution of this order sorting GS (OSGS) is matched to the FPI's FSR. The OSGS will cut out slices from the FPI ring system on the detector, where single FPI transmission orders are isolated (see Fig. 2d). The widths of these slices are given by the OSGS's ILF, i.e. its slit width. The result is a variable étendue across the FPI spectrograph's focal plane, generally decreasing with increasing distance to the center of the ring system (i.e. increasing

incidence angle $\alpha$). In the following, an approximate quantification of the étendue $E_\mathrm{H,FSG}$ of the FPI spectrograph with grating OSM is derived. We thereby regard the area on the detector, where the rings of equal FPI transmission are approximately



parallel to the grating dispersion dimension (e.g. a bit above the centre of the FPI ring system). There, the grating dispersion and the FPI dispersion are approximately perpendicular (see Fig. 2d). Light from within a wavelength interval $\delta\lambda_{\mathrm{FPI}}$ covers the area $A_{\mathrm{H,FSG}}$ on the detector. For 1:1 imaging, its horizontal extent (in grating dispersion direction) is given by the OSGS slit width $w_{\mathrm{S}}$.

The vertical extent of $A_{\mathrm{H,FSG}}$ can again be approximated by the radial change of the detector location upon a shift of the transmission peak at $\lambda_{\mathrm{m}}$ by $\delta\lambda_{\mathrm{FPI}}$. Thereby, $A_{\mathrm{H,FSG}}$ becomes a function of the imaging focal length $f_2 = f_1$ and the angle range $\Delta\alpha$ required for tuning the FPI by $\delta\lambda_{\mathrm{FPI}}$. Again, linearising Eq. 4 yields:

$$\Delta\alpha \approx \frac{\delta\lambda_{\mathrm{FPI}}}{-\lambda_{\mathrm{m}}\,\alpha} \tag{E1}$$

This approximation should be fine for $\alpha > \sqrt{2\frac{\delta\lambda_{\mathrm{FPI}}}{\lambda_{\mathrm{m}}^0}}$ (see Eq. (D2)), where the FPI angular dispersion does not diverge. The product of $\Delta\alpha$ and the imaging focal length $f_2$ is then the vertical extent of $A_{\mathrm{H,FSG}}$:

$$A_{\mathrm{H,FSG}} \approx w_{\mathrm{S}}\,\frac{f_2}{\alpha}\,\frac{\delta\lambda_{\mathrm{FPI}}}{\lambda_{\mathrm{m}}} \tag{E2}$$

The solid angle of a light beam reaching a detector spot is again given by the imaging optics F-number (which should be matched to the OSGS's F-number):

$$\Omega_{\mathrm{H,FSG}} \approx \frac{\pi}{4\,F^2} \tag{E3}$$

Finally, we obtain the étendue of the FPI spectrograph with grating OSM:

$$E_{\mathrm{H,FSG}} \approx A_{\mathrm{H,FSG}}\cdot\Omega_{\mathrm{H,FSG}} \approx \frac{\pi}{4\,F^2}\,w_{\mathrm{S}}\,\frac{f_2}{\alpha}\,\frac{\delta\lambda_{\mathrm{FPI}}}{\lambda_{\mathrm{m}}} = \frac{\pi}{4\,f_2}\,\frac{w_{\mathrm{S}}\,b_{\mathrm{FPI}}^2}{\alpha}\,\frac{\delta\lambda_{\mathrm{FPI}}}{\lambda_{\mathrm{m}}} \approx \frac{w_{\mathrm{S}}}{2\,\pi\,f_2\,\alpha}\,E_{\mathrm{H,FPI}} \tag{E4}$$

**Appendix F: On the implementation of the interferometric OSM**

In principle, the FPI can be used together with a bandpass filter with a transmission FWHM of the FSR of the FPI. In the UV around 300 nm, for a FPI with resolving power of $R = 150000$ and a finesse of $\mathcal{F} = 100$ the bandpass FWHM should be around 0.2 nm. Such filters with a transmission of about 25-35 % are available (see e.g. Klanner et al., 2021).

Alternatively, an additional FPI with lower resolving power can be used to increase the effective FSR and thereby the required FWHM of the interference filter bandpass (as e.g. in Mack et al., 1963). The étendue will then still be limited by the FPI with the highest resolving power (see Eq. (25)).

The resulting ring shaped raw spectra are translated into linear spectra by coadding the intensity of all pixels with the same distance to the center of the ring system. Alternatively a hardware based circle to line converter (as e.g. proposed in Hays, 1990) can be used.

*Author contributions.* JK conceptualised and performed the theoretical study, built the prototype and wrote the draft of the manuscript. All coauthors substantially contributed to the refinement of the study and revised the manuscript in several rounds.



610 *Competing interests.* The authors declare that they have no conflict of interest.

*Acknowledgements.* We would like to thank SLS Optics Ltd. for sharing their expertise in designing and manufacturing etalons. The authors also like to thank the German Science Foundation (DFG) for partial funding through the project PL 193/23-1.





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
