# Peer review of "Mobile and high spectral resolution Fabry Pérot interferometer spectrographs for atmospheric remote sensing"

_Atmospheric Measurement Techniques, 2021_

## Author Comment (AC1)

***Reply on RC1: 'Comment on amt-2021-133' by Anonymous Referee #3***

*We thank the reviewer for reading the manuscript and for the feedback. From the minor comments and questions concerning the manuscript we assume that the review recommends publication in AMT. In the following we answer (italic font) the comments (cited in roman font) and indicate manuscript modifications (bold font).*

The authors advocate for the increased use of Fabry Perot Interferometers (FPI) in trace gas remote sensing. Conceptual comparisons are made between FPI and grating spectrometers (GS) that are more widely used for trace gas remote sensing, with some notional quantification of spectral resolution, signal to noise ratio (SNR) and mass/volume. The authors emphasize the mobility of such an instrument. The case is made that though the SNR is 1%-10% that of a GS, the spectral resolving power can be 30 times greater. A prototype FPI is presented with images visually comparing data and calculations.

*Many thanks for this positive assessment. This summary is correct, except for the fact that the resolving power of FPI instruments can be 250 times (not 30 times) larger than that of a GS for 1%-10% the SNR and the same instrument size (see e.g. 'Abstract','Sect. 3.2.3', 'Conclusions').*

General Comments:

The authors attempt a thorough presentation of important FPI features related to the optics in order to compare the resolving power of the GS and FPI. However, for a mobile instrument no mention is made of temperature control/stability of the FPI needed for high resolution systems which would presumably have a significant dependence on the stability of the refractive index of the optics.

*Temperature stability indeed is an important point concerning mobile measurements. Through the use of air spaced etalons with low thermal expansion glass as spacers (see l. 168 of the original manuscript) temperature stability is given for most applications without requiring active control.*
*The temperature expansion coefficient of low expansion glass is about $10^{-8}$/K (see e.g. schott.com). A temperature change by $dT = 10K$ represents an extreme value during an exposure of a spectrum. According to Eq. (1) and (2) the relative spectral shift $d\lambda$ of the FPI transmission spectrum equals the relative change in the separation of the reflective surfaces $dd$: $dd/d = d\lambda/\lambda$. Thus, for the assumed resolving power ($\lambda/d\lambda=150000$) the assumed extreme temperature change of 10K would shift the FPI transmission spectrum only by 1/66 ILF width.*
*Regarding changes in the index of refraction of the gas between the plates: The refractive index of air is mostly dependent on the number density of air molecules between the two reflective surfaces. By hermetically sealing the volume between the two glass surfaces (offered as an option by many manufacturers, see e.g. slsoptics.com) the number density of air molecules remains constant.*
*Despite this very low temperature sensitivity there may be applications (maybe air-borne) where active temperature stabilisation is needed. In this case, figures would be the same as for GSs, which in most cases need active temperature stabilisation in atmospheric remote sensing applications (see e.g. Platt and Stutz, 2008).*
*Further, the temperature impact on FPIs, as well as that on the simple optical setup, can be captured by models of the instrument transmission. This is much more difficult for GSs, since there the temperature also significantly affects the rather non-linear imaging of the slit.*

*We further stressed this very important advantage of FPI spectrographs in the revised manuscript (l. 181 of the original manuscript):*

**Generally, a static setup (without moving parts) has a high mechanical stability and low maintenance requirements. This is demonstrated by moderate resolution GS applications. Spectrographs using FPIs implemented with low thermal expansion glass (linear expansion coefficient $\gamma = 10^{-8}$/K) spacers further yield superior thermal stability. From Eqs. (1) and (2) follows that $d\lambda / \lambda \sim \gamma \, dT$. A rather extreme temperature change of 10K then induces a shift of the transmission spectrum by $10^{-7} \lambda$. Even for a high resolving power of $10^5$ this would hardly have an effect on the measurement. The issue of potentially varying air density within the etalon impacting the refractive index is solved by hermetically sealing the etalon. Further, the temperature impacts on FPIs, as well as that on the simple optics, can be accounted for in models of the instrument transmission. This is much more difficult for GSs, since there temperature also significantly affects the rather non-linear imaging of the slit. Thus, while GSs often require active temperature stabilisation (see Platt and Stutz, 2008), for FPI spectrographs it might be redundant for most applications. This substantially enhances their mobility through a simpler and smaller setup with lower power consumption.**

The authors fail to mention FPIs are commonly used for observing airglow and auroras. Fiber Bragg-grating versions are also used in receivers for in-elastic scattering lidars.

*In our discussion of 'Atmospheric trace gas remote sensing with high spectral resolution' (Sect. 1.2) we focus on absorption measurements to quantify atmospheric trace gases, mostly performed with spectrograph instruments. Further, we state that our calculations are not valid for the spectroscopy of sharp (e.g. atomic) line emitters (see l. 294-297 of the original manuscript). A comprehensive range of FPI applications (certainly encompassing airglow emissions and many more applications) are listed and discussed in detail in the literature, for instance in Vaughan, 1989 (quoted in the manuscript).*
*Further, it is not clear to us, how a fibre Bragg-grating could be used to implement a high resolution spectrograph.*
*Thus, we prefer not to explicitly mention FPI-based airglow and aurora observations and LIDAR applications in the context of our study.*

The authors do not mention grating-prism instruments (aka grisms) that should offer improvements in resolving power over GS with a small increase in volume/mass.

*We do not see a substantial advantage of a grating-prism combination for the application of high resolution absorption spectroscopy of atmospheric trace gases. Jacquinot (1954) showed that across all relevant wavelength ranges the dispersion of an optimised grating is always considerably larger than that of a prism. Thus, the dispersion of the grating and the prism in an appropriate combination might sum up, increasing the total dispersion by considerably less than a factor of two. The disturbing effects of introducing an additional prism to the light path (temperature dependence, spectrograph stray light, etc.) might finally outweigh the modest increase in resolving power.*
*Similar arguments hold for immersion gratings. Such small impacts on the resolving power ($<<100$, which is about the difference between grating and FPI) are accounted for by the factor $\kappa$ (introduced in l. 332-336 and l.534 of the original manuscript) in our calculations.*

*Note that in the introduction we included the following statement (l. 31 original manuscript):*

*For resolving powers higher than a few thousands, Jacquinot (1954, 1960) showed that the FPI exhibits a fundamental luminosity (or light throughput) advantage over gratings*, **which, in turn, outperform prisms in all relevant wavelength ranges**.

The estimation of SNR for the FPI and GS is somewhat through, but the following discussion concerning level of detection is more vague. Do you have a reference for the statement in line 436 "…the sensitivity increase will be almost linear to the increase in spectral resolution…"

*We thank the reviewer for the compliments on our SNR estimations.*
*We agree that the relation between sensitivity and spectral resolution needs to be better explained. If the width $\delta\lambda_{line}$ of an absorption line is much smaller than the ILF width $\delta\lambda_{ILF}$ of an observing spectrograph, the peak absorption of the apparent (measured) line is reduced by about the factor $\delta\lambda_{line}/\delta\lambda_{ILF}$ with respect to the true absorption of the line. This 'dilution' of a line (i.e. the reduction of peak absorption due to not fully resolving a line) within the ILF determines the sensitivity i.e. the effective absorption cross section $\bar{\sigma}$ of the gas to be measured. Thus, until the ILF width reaches the width of the absorption line, we expect an approximately linear relation of sensitivity and spectral resolution. In the added Appendix B, we include an explanation and illustration (Figure B1) of the sensitivity – spectral resolution relation for a Voigt-shaped absorption line and ILF modelled by a higher order Gaussian. The Appendix is added in l. 529 and referenced in l. 293 and l. 437 of the original manuscript:*

**Appendix B: Relation between sensitivity and spectral resolution**

**Here we want to motivate that the sensitivity of an absorption measurement with a spectrograph is in most cases strongly dependent on its spectral resolution. The sensitivity can be approximately quantified by the peak effective absorption cross section $\bar{\sigma}$ of a gas measured by an instrument with ILF H:**

$$\bar{\sigma}(\lambda) = \frac{\tau(\lambda)}{S(\lambda)} = S(\lambda)^{-1} log \frac{I_0(\lambda) \otimes H(\lambda)}{I_0(\lambda) exp(-\sigma(\lambda)S(\lambda)) \otimes H(\lambda)}$$

**Here σ denotes the high resolution absorption cross section, S the column density of the gas and the operator $\otimes$ represents the spectral convolution. The absorption of an isolated and sharp absorption line (see e.g. OH absorption cross section in Fig. 1) is reduced within the ILF of a spectrograph as long as its spectral resolution is lower than the width of the absorption line. In this case, increasing the spectral resolution results in a close to linear increase in sensitivity. This is illustrated by a simple example in Fig. B1, where we assume a 3 pm wide, Voigt-shaped absorption line and ILFs of different width modeled by 6th order Gaussian curves.**

[Figure]

**Figure B1: The absorption of a sharp line is 'diluted' throughout the ILF H of the observing spectrograph. For ILF widths δλ that are much larger than the width of the absorption line, the measured absorption signal (peak optical density, i.e. peak effective absorption cross section $\overline{\sigma}$) increases approximately linearly with spectral resolution (i.e. with 1/δλ). For this visualisation the ILF was modelled with a higher order Gaussian (6th order) and a Voigt profile was assumed for the absorption line.**

Line 440: In this discussion of reducing the temporal resolution of the FPI are you also reducing the temporal resolution of the "moderate (spectral) resolution DOAS measurement?" Using this dramatic approach to increase the SNR will only work for certain investigations that can afford such coarse knowledge. Such a direct means of SNR increase does not seem applicable to air-borne based instruments.

*In this discussion we assumed the temporal resolution of the moderate resolution DOAS measurement not to be reduced. We clarify this by adding the note "**(with 30s exposure time)**" in brackets to the statement (l.442 original manuscript).*
*As noted in Sect. 3.2 (l. 285 original manuscript, particularly Eq. (14)) the absorption cross section of the gas, the spectral resolution and the light throughput of the instrument, the radiance, and the time constant of the process to be studied (determining the exposure time) determine whether a process can be studied with a certain spectrograph. This clearly implies that there are also processes that cannot be studied (as for instance tropospheric OH measurements with GSs and scattered sunlight as a light source). We do not claim that any arbitrary atmospheric process can be studied with FPI spectrographs.*
*On the other hand there are air-borne platforms that are able to stay in the air for several hours, i.e. several integration times of a FPI spectrograph with the same photon SNR as a moderate resolution GS (i.e. with 0.004 the detection limit for many gases). Depending on the intended observation this might still be sufficient temporal resolution. For comparison, a 1000 times larger GS reaching the same photon SNR with the same spectral resolution as the FPI-instrument would need an integration time of 5 days (compare light throughput of FPI spectrograph and high resolution spectrograph in Table 1 in the manuscript) and would thus be applicable to much less air-borne atmospheric observations.*

There is no dispute that FPIs could be used more widely for trace gas detection. The practical question to answer is whether or not they can achieve the suggested detection capabilities in practice for the mobile scenarios. The prototype instrument is interesting and demonstrated quantitative performance characteristics should be included, in addition to the visual side-by-side comparison of measurements and model results shown in Fig. 5.

*We agree that a quantitative assessment of prototype instruments would be a valuable addition. However, as noted in the manuscript (l. 486-487) it would go beyond the scope of this work. This manuscript intends to provide a thorough theoretical understanding in order to motivate the use of the proposed FPI spectrographs as well as application specific prototype design and construction. The short paragraph 4.2 illustrates that FPI spectrographs can be designed in a straightforward manner.*

Minor Comments:

Line 99 is missing a comma before *water*.

Line 416 should delete "by" to then read "…throughput is about…"

*We performed the suggested changes and thank the reviewer very much.*

---

## Author Comment (AC2)

*Reply on RC2: 'Comment on amt-2021-133' by Anonymous Referee #2*

*We thank the reviewer for the positive assessment of our work and the recommendation for publication in AMT. In the following we answer (italic font) the minor comments (cited in roman font) and indicate manuscript modifications (bold font).*

This is an excellent paper describing the advantages of using Fabry-Perots interferometers (FPI) in compact spectrographs for remote sensing measurements in the atmosphere. The authors emphasize that FPI-based instruments provide higher resolving power in comparison to conventional grating spectrographs (GS) of similar physical dimensions. The results are convincing and supported by an elegant prototype demonstration. The paper is very well written and certainly suits the scope of AMT. I have only minor suggestions to be addressed in the manuscript:

1. The authors limit their consideration to the geometrical dimensions of the instrument. However, more factors can and should be considered in the development of the mobile optical instrument. I would name mechanical and thermal stability, cost of production, and maintenance requirements. The paper would benefit if some of these points could be addressed by the authors.

*We agree with the reviewer on the point that mechanical and thermal stability and low maintenance requirements are important aspects of mobility. In the abstract and in Sect. 1.3 (l. 115 original manuscript) we define 'mobility' as 'compact and stable'. 'Compact' refers to the geometrical dimensions of the instrument and 'stable' to mechanical/thermal stability that induces low maintenance requirements. Our argument is that FPI spectrographs maintain the high mobility of moderate resolution GS (which have proven their mobility, see Sect. 1.3) and, moreover, that modern FPI spectrograph designs exhibit excellent thermal stability, likely superior to most commercial GS designs. We refer to our answers to RC1 and the addition of the discussion of thermal stability to the revised manuscript (l. 181 of the original manuscript):*

**Generally, a static setup (without moving parts) has a high mechanical stability and low maintenance requirements. This is demonstrated by moderate resolution GS applications. Spectrographs using FPIs implemented with low thermal expansion glass (linear expansion coefficient $\gamma = 10^{-8}$/K) spacers further yield superior thermal stability. From Eqs. (1) and (2) follows that $d\lambda / \lambda \sim \gamma \, dT$. A rather extreme temperature change of 10K then induces a shift of the transmission spectrum by $10^{-7}$ $\lambda$. Even for a high resolving power of $10^5$ this would hardly have an effect on the measurement. The issue of potentially varying air density within the etalon impacting the refractive index is solved by hermetically sealing the etalon. Further, the temperature impacts on FPIs, as well as that on the simple optics, can be accounted for in models of the instrument transmission. This is much more difficult for GSs, since there temperature also significantly affects the rather non-linear imaging of the slit. Thus, while GSs often require active temperature stabilisation (see Platt and Stutz, 2008), for FPI spectrographs it might be redundant for most applications. This substantially enhances their mobility through a simpler and smaller setup with lower power consumption.**

2. Despite being essentially different in principle, laser spectroscopy offers high resolution and selective detection of trace gases and recent developments also show that the laser-based solutions can also be made compact and energy-efficient. The presented development could be put in the context of state-of-the-art mobile/compact laser systems.

*We thank the reviewer for this valid remark. Laser light sources do exhibit high radiance and high spectral resolution. Therefore one might conclude that for active absorption spectroscopy (i.e. not using natural light sources like the sun) laser-based instruments should be superior to any other spectrometer. However, in practice lasers are frequently complicated, bulky, and expensive devices with limited spectral tuning range. Simple and compact designs, like diode lasers have only been available for very limited spectral ranges. It is probably because of these limitations that during the last decades their role in atmospheric remote sensing (i.e. open path measurements) was smaller than expected. For most applications, laser systems are not limited by photon statistics, which makes a comparison with a spectrograph system quite complex. A sound discussion of laser systems, potential future development and the comparison to techniques using incoherent light sources would be interesting, but quite a comprehensive task. It could be the topic of an independent review on atmospheric trace gas remote sensing. Here we limit our study to spectrograph measurements.*
*Despite of atmospheric remote sensing using laser technology being an emerging field (see e.g. Fiddler et al. 2009), we are convinced that it is worthwhile to further develop simple techniques using incoherent and natural light sources to study atmospheric trace gases.*

*(Marc N. Fiddler, Israel Begashaw, Matthew A. Mickens, Michael S. Collingwood, Zerihun Assefa, and Solomon Bililign: Laser Spectroscopy for Atmospheric and Environmental Sensing, Sensors 9, 10447-10512; doi:10.3390/s91210447,2009)*

3. Line 448, the discussion on isotope detection is somewhat vague. Could you add what isotopes and in which compound you are aiming to detect?

*In the Introduction (Sect 1.2, l. 102-104) we mentioned the example of water vapour isotope measurements, which are currently performed with lower resolution. The sensitivity could be substantially enhanced through a higher spectral resolution.*
*In the UV and visible wavelength range the difference in absorption cross sections of isotopologues of small trace gas molecules is often on a sub nm scale that cannot be resolved with moderate spectral resolution. High spectral resolution with comparable SNR might thus allow separating the absorption of different trace gas isotopologues as long as their abundance is high enough. For instance, in addition to water vapour isotopologues the measurement of the $^{34}SO_2$ in volcanic plumes could be possible.*

*We replace the sentence (l.448 original manuscript)*

*Consequently, also the feasibility of distinguishing trace gas isotopes is strongly enhanced and line broadening effects could add valuable information to retrievals of vertical atmospheric trace gas distributions.*

*by*

**Consequently, also line broadening effects could add valuable information to retrievals of vertical atmospheric trace gas distributions and the feasibility of distinguishing trace gas isotopologues is strongly improved. Besides improving water isotopologue quantifications (e.g. Frankenberg et al. 2009) the separation of $^{34}SO_2$ and $^{32}SO_2$ in volcanic emissions could be possible through the differences in the absorption cross section, which are on a sub nm scale (Danielache et al., 2008) and thus impossible to resolve with moderate spectral resolution.**

*With the additional reference:*

**Danielache, S. O., Eskebjerg, C., Johnson, M. S., Ueno, Y., and Yoshida, N.: High-precision spectroscopy of 32S, 33S, and 34S sulfur dioxide: Ultraviolet absorption cross sections and isotope effects, Journal of Geophysical Research, 113,https://doi.org/10.1029/2007jd009695, 2008.**

4. Minor technical comment: Line 439 "... will be by **a factor of ... 0.04 lower** than that of ... " Please check if it is a correct statement.

*We thank the reviewer for this remark. For clarification, we replaced "… will be by a factor of … 0.04 lower than that of common DOAS measurements with moderate spectral resolution" (l. 439 original manuscript) by*

**… will be reduced by a factor of … 0.04 compared to that of common, moderate spectral resolution DOAS measurements.**

Again, this is an excellent paper that deserves publication!